# The shift of phosphorus transfers in global fisheries and aquaculture

Yuanyuan Huang [1,2]*, Phillipe Ciais [1], Daniel S. Goll [1,3], Jordi Sardans[4,5], Josep Peñuelas [4,5], Fabio Cresto-Aleina[1] & Haicheng Zhang[1,6]

Global fish production (capture and aquaculture) has increased quickly, which has altered global flows of phosphorus (P). Here we show that in 2016, $2.04^{3.09}_{1.59}$ Tg P yr$^{-1}$ (mean and interquartile range) was applied in aquaculture to increase fish production; while $1.10^{1.14}_{1.04}$ Tg P yr$^{-1}$ was removed from aquatic systems by fish harvesting. Between 1950 and 1986, P from fish production went from aquatic towards the land-human systems. This landward P peaked at 0.54 Tg P yr$^{-1}$, representing a large but overlooked P flux that might benefit land activities under P scarcity. After 1986, the landward P flux decreased significantly, and became negative around 2004, meaning that humans spend more P to produce fish than harvest P in fish capture. An idealized pathway to return to the balanced anthropogenic P flow would require the mean phosphorus use efficiency (the ratio of harvested to input P) of aquaculture to be increased from a current value of 20% to at least 48% by 2050 — a big challenge.

[1] Laboratoire des Sciences du Climat et de l'Environnement, LSCE/IPSL, CEA-CNRS-UVSQ, Université Paris-Saclay, 91191 Gif-sur-Yvette, France. [2] Commonwealth Scientific and Industrial Research Organisation, Aspendale, 3195 Victoria, Australia. [3] Department of Geography, University of Augsburg, Augsburg, Germany. [4] CSIC, Global Ecology Unit CREAF-CSIC-UAB, 08913 Bellaterra, Catalonia, Spain. [5] CREAF, 08913 Cerdanyola del Vallès, Catalonia, Spain. [6] Department Geoscience, Environment and Society, Université Libre de Bruxelles, 1050 Bruxelles, Belgium. *email: yuanyuanhuang2011@gmail.com

One big challenge humanity faces today is the phosphorus (P) dilemma. Phosphorus (P) is an essential element for all forms of life on Earth. The rapid rise of human demand for food has quadrupled P inputs into the biosphere since the preindustrial time. These inputs have primarily occurred through mineral P fertilizer addition[1,2]. In the long term, P scarcity may threaten global food production because the ore deposits from which P fertilizer is made are depleting and likely to be exhausted in the future[1,3]. On the other hand, today, the distribution of P is highly uneven[4], and regional surpluses of P in croplands, inland waters, and coastal seas are not uncommon[1,4–8]. Excessive P in inland and coastal waters has been widely recognized as the dominant driver of eutrophication, which degrades water quality, decreases biodiversity, alters ecosystem dynamics, and results in dead zones[7–11]. These widespread adverse consequences of eutrophication triggered recent worries that the planetary boundary estimated to be a load of $0.89\,kg\,P\,yr^{-1}$ per capita or $6.2\,Tg\,P\,yr^{-1}$ in total to the ocean (mostly from the land) would be exceeded, i.e., its safe global operating space would be transgressed[12–14].

Food production is the largest driver of large-scale anthropogenic P release into aquatic ecosystems, and recent global-scale P budgeting has focused on agriculture[6,15,16]. In total, 82.4% of phosphate fertilizers goes to cropland and pasture. The high rate of manure and mineral P fertilizer application to agriculture results in around a half of the P not being taken up by plants, increasing the risk of P being transferred into aquatic ecosystems through erosion, runoff, and leaching[6,17–19]. Phosphorus applied to cultivated soils in livestock manure exceeds the global mineral P fertilizer use[17], and a third of the P transferred into freshwater is attributable to the livestock sector[18]. Human waste processing or disposal, and the use of detergents also release P from consumed food products into inland and coastal waters. As a result, current mitigation strategies and technological innovations concentrate on recycling and better management of soil, crop, and livestock P flows, and improved rates of recovery P from wastewater[19,20].

The global fishery is an overlooked food production subsector that is critical in land and aquatic nutrient flows[21–24]. Finfish, crustaceans, and mollusks, hereafter generalized as fish, contribute substantially to the global animal protein supply for humans (~17% in 2013). The annual global fish harvest, including both capture and aquaculture, has increased from 19 Tg in 1950 to 169 Tg in 2016[25]. Fish harvest returns nutrient to land–human systems but is currently an underrepresented aspect of anthropogenic P fluxes globally and regionally. Aquaculture has been the fastest-growing sector of food production over the past decade[25] and is expected to expand further[22,26]. However, its dependence on wild fish and crop livestock for feeds, the use of water and land resources, and other environmental impacts on aquatic ecosystems has cast doubt on the environmental sustainability of aquaculture[21,22,27–30], along with growing sustainability concerns on other food production sectors. As with the husbandry of livestock, aquaculture production relies on external supply of P either directly through feeds (e.g., for carnivorous fish) or through fertilizers that enhance the primary productivity of aquatic ecosystems (e.g., for herbivorous and omnivorous species). Phosphorus that is not harvested might end up in inland and coastal waters and result in eutrophication[9,31,32]. More than 90% of fish farming occurs in Asia where P-use efficiency (PUE) defined as the ratio of harvested to input P for a given farming system is generally low. For example, aquaculture PUE ranges from 8.7% to 21.2% in China[32], indicating large loss rates to the environment.

The global fishery production system is highly diverse with respect to harvested fish species (528 in 2014)[33], fates of harvested fish, intensities of culture practice (e.g., extensive, semi-, and intensive), aquatic ecosystems (e.g., freshwater, brackish water, or marine), background environment (e.g., water chemistry), rearing facilities (e.g., ponds, cages, pens, tanks, or raceways), number of species (e.g., polyculture vs. monoculture), and the socioeconomic status of fish farmers. As a result, a data-driven quantification of anthropogenic impacts of fishery production on large-scale P budgets is challenging. This study fills the gap through providing a data-driven quantification of the net impact of the global fishery on P flows, improving basic understandings of P biogeochemical cycles, and providing support to identify the P management targets. We reconstruct global P budgets driven by fishery production by an extensive compilation of data coming from global fishery production databases (see the "Methods" section, Supplementary Fig. 2), fish stoichiometry from a whole-body P concentration (including bone) dataset with 1164 records for 224 fish species (see Methods, and Supplementary Figs. 3 and 4), and aquaculture farm system-level PUE estimates for a representative range of aquaculture farm types with 168 entries (see Methods, Supplementary Fig. 5). In addition, we also compile a P-retention efficiency (PRE) database across controlled feeding experiments (Supplementary Fig. 6). We define PUE as P harvested via fish biomass divided by P input via feed and fertilizer. PUEs are calculated at the farm level. Fish feed includes commercially manufactured compound aquafeeds with different additives, farm-made feeds from crop–livestock products and/or by-products, and fish with low economic value or by-products recycled from aquaculture industry etc. Fertilizers cover inorganic chemical fertilizers and manures from a variety of sources, such as poultry, ruminant waste, swine waste, and human excreta. PRE is the fraction of P that is recovered in harvested fish biomass per feed P intake from controlled feeding experiments (see Methods for details).

We show the global net anthropogenic P flow between land–human and aquatic systems through fisheries and aquaculture (Fig. 1; see also Supplementary Fig. 1, Supplementary methods: Boundary conditions). Here aquatic systems include both marine and freshwater systems. The net P flow consists of two major gross fluxes (opposite in directions): P flows extracted from aquatic ecosystems by fish harvest (P-harvest) and P input flows to aquatic systems in the form of feed and fertilizers (P-input). We use P-harvest minus P-input, that is, P-net, to quantify the net P transfer between aquatic and land–human systems driven by the global demand for fish meat. When P-net is positive, P moves out of aquatic ecosystems and is a short-term gain for land–human systems. On a longer term, fish P waste after consumption or during fish food processing can be dumped to landfills, recycled to livestock and croplands, or recycled back to rivers and returned to aquatic systems. Conversely, when the balance is negative, it means that humans add to aquatic ecosystems more P for fish production than they retrieve in harvesting wild and cultivated fish. Annual P-harvest (mean and its range, 1950–2016) is estimated from combining fish production with the whole-body P concentration for different fish species (see Methods and caption to Fig. 2). Annual P-input is derived from P-harvest and aquaculture system-level PUE (see Methods and caption to Fig. 2). PRE is treated as a reference PUE given by current technology and the PUE level that the global fishery management could potentially reach. Tracking of the fate of harvested P, the relative form of P losses in aquaculture (e.g., excrements, uneaten feed, or by-products), and the location of P losses (e.g., during transport, in hatcheries, or during fish processing) through time are not the major focus of this study (see Fig. 1, Methods and Supplementary Methods for details) due to limited reliable global datasets.

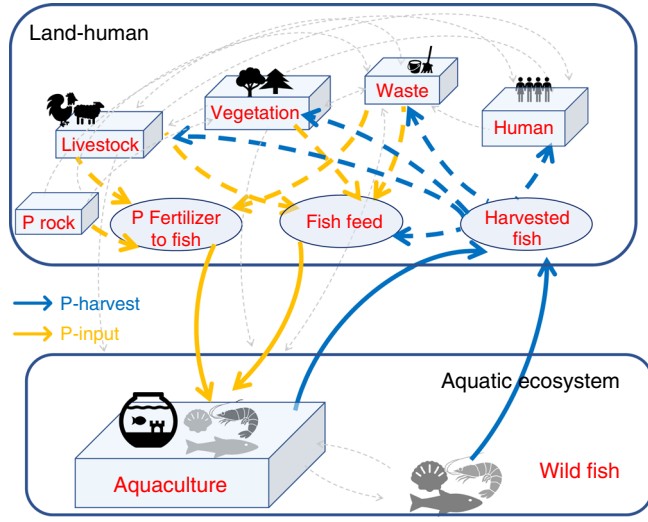

**Fig. 1 Overview of the global fishery P flows.** Harvested fish transport P from aquatic ecosystems to land (blue arrows, Methods: Data, P budget), while fish farming requires P input primarily through fish feed and P fertilizer (orange arrows, Methods: Data, P budget). The global fishery P transfers involve complex interactions among terrestrial vegetation, livestock, and human society and waste managements (dashed gray arrows, see Supplementary Fig. 1 for more details). We focus on the fishery-caused major perturbation of P flows between aquatic ecosystems and land. Aquatic systems here include both marine and freshwater systems. We adopt a land or human-centric viewpoint. We call external P that goes directly into the aquatic environment through fisheries and aquaculture, P-input, and P that moves out of the aquatic ecosystems through harvested fish, P-harvest. We use P-net that is the difference between P-harvest and P-input, to quantify the net P budget between aquatic ecosystems and land. Details of the boundary conditions are provided in the Supplementary Information. Animal and tree silhouettes are from Microsoft PowerPoint (office 365) icon.

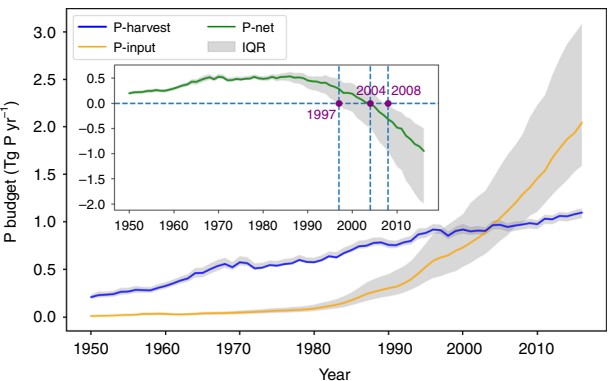

**Fig. 2 Global fishery phosphorus balance from 1950 to 2016.** The blue line shows the average P-harvest flux from fish. The orange line represents the average P-input into aquatic ecosystems through aquaculture. The inset figure with the green line is the net P-net, i.e., P-harvest minus P-input. Light-gray shading indicates interquartile ranges (IQR, the 75th and 25th percentiles) calculated from 1000 estimates with randomly sampled fish P concentration, fish biomass within 50% percentile uncertainty for P-harvest, and culture-system level P-use efficiency for P-input. Units are Tg P $yr^{-1}$. Source data are provided as a Source Data file.

## Results and discussion

**Historical and future quantifications.** The global P-harvest (wild + aquaculture) increased from $0.21_{0.19}^{0.24}$ (mean and

interquartile range) in 1950 to $1.10_{1.04}^{1.14}$ Tg P $yr^{-1}$ in 2016; at the same time, P-input (aquaculture feed and fertilizer) grew from $0.01_{0.007}^{0.015}$ to $2.04_{1.59}^{3.09}$ Tg P $yr^{-1}$ (Fig. 2, Supplementary Figs. 7 and 8). Our global estimate of P-input for 2010 (1.46 Tg P) inferred from farm-level PUE data is close to the value from the World-Fish database (1.11 Tg P) obtained by multiplying the estimated farming area by the parameter representing per unit area nutrient and feed input[28]. P-harvest estimated here is the largest pathway that transfers P from aquatic ecosystems to land, compared with currently known pathways, i.e., 0.0056 Tg P $yr^{-1}$ from anadromous (migratory) fish[34], 0.099 Tg P $yr^{-1}$ from seabird colonies[35], and 0.16 Tg P $yr^{-1}$ from sea salt deposition[36]. In all, 99% of P-harvest came from wild capture (mostly marine capture) in 1950, and this share decreased to 62% in 2016 (Supplementary Figs. 7 and 8). Because wild fish capture dominated fish production in the early decades of the record (Supplementary Fig. 2), P-harvest outweighed P-input, resulting in a net removal of P (positive P-net) from aquatic ecosystems, which reached a maximum at $0.54$ Tg P $yr^{-1}$ in $1986_{1970}^{1987}$. Aquaculture took off dramatically in the 1980s accompanied by increased P-input, while PUE did not increase at the same rate as P-input. This expansion of aquaculture led to a net flux of P from land–human systems to aquatic ecosystems—a negative P-net for the global fishery sector. The turning point from a positive to a negative global P-net occurred around $2004_{1997}^{2008}$. Today, the global P-net is clearly negative and amounts to $-0.95_{-1.99}^{-0.50}$ Tg P $yr^{-1}$. In comparison, leaching, runoff, and erosion losses of fertilizer P from croplands[37] to freshwater are reported to be 0.6 Tg P $yr^{-1}$ from Mekonnen and Hoekstra [37] over 2002–2010, and Lun et al.[6] report a larger P loss rate of 3.7 Tg P $yr^{-1}$ from croplands to water bodies through runoff over the same period. For reference[4–22], Tg P $yr^{-1}$ are transported from rivers to oceans (Ref. [38]: 4 Tg P $yr^{-1}$; Ref. [39]: 9 Tg P $yr^{-1}$; Ref. [40]: 22 Tg P $yr^{-1}$). The role of the global fishery sector in loading P from land to aquatic systems thus represents an important component of anthropogenic P transfers. Freshwater aquaculture contributes to most (84–94%) of the P-input. The share of marine aquaculture increases through time (Supplementary Fig. 8). Within the freshwater aquaculture, most of P-input ends up in raising finfish (95–100%), and the share from raising crustaceans slightly increases with time, to reach 5.31% in 2010. In the marine aquaculture, most of P-input (>90%) ends up in finfish aquaculture during 1950–1970; however, after 1990, around 50% of marine aquaculture P-input goes into raising crustacean species (Supplementary Fig. 8).

At the continental scale, Asia has been driving the global shift to a negative P-net, with P-net remaining positive in Europe, America, Africa, and Oceania, where P-harvest continued to exceed P-input (1950–2016, Fig. 3). P-input first outweighed P-harvest around year $1988_{1974}^{>2016}$ in South Asia, by $1990_{1982}^{2008}$ in East Asia, $2005_{1985}^{>2016}$ in Southeast Asia, and $2016_{1998}^{>2016}$ in West Asia. In the year 2016, Asia alone contributed a negative P-net of $-1.13$ Tg P $yr^{-1}$ and the rest of the world a small positive P-net of 0.18 Tg P $yr^{-1}$.

Country-wise, enhancements in P-input from developing Asian countries, and to a lesser extent, decreases in P-harvest from wild fish captures in some developed countries, contributed to the overall negative trend in P-net, which crossed the zero line downward after $2004_{1997}^{2008}$ (Fig. 4, Supplementary Figs. 12, 13, and 14). Between the recent decade (2007–2016) and the earlier decade when P-net was approaching its positive maximum (1980s), P-input in aquaculture increased by a factor of nine. Developing Asian countries contributed most of the increased P-input, with China accounting for 60% of the global increase, India 10%, Indonesia 6%, Vietnam 6%, Bangladesh 3%, and

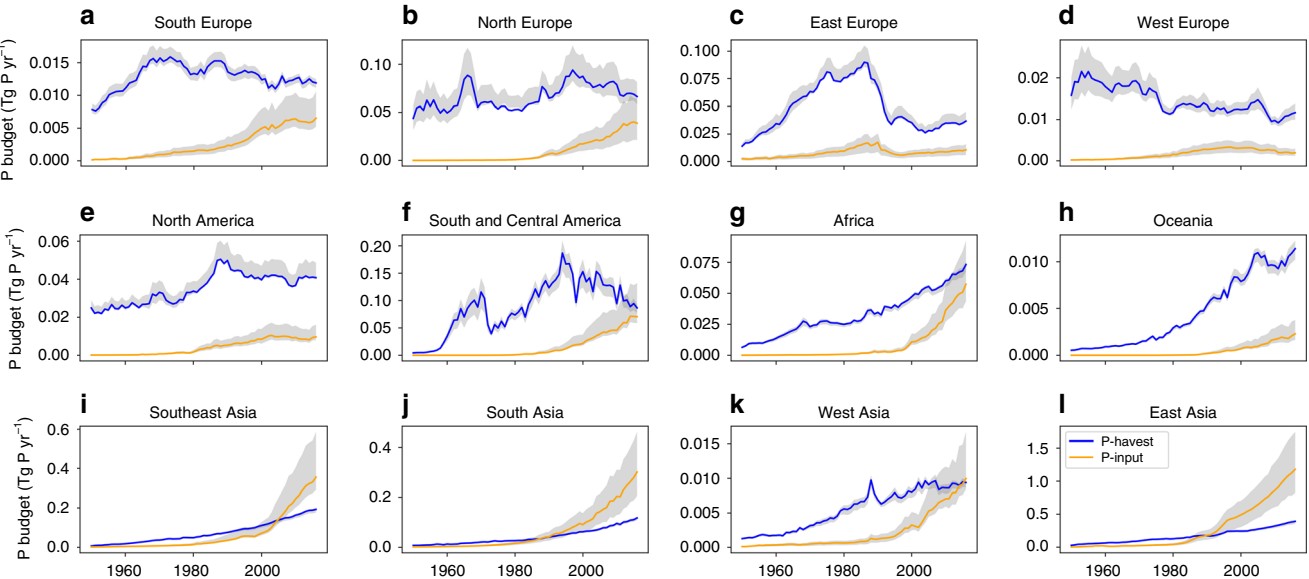

**Fig. 3 Continental-scale fishery phosphorus balance from 1950 to 2016.** Continent designation is based on FishStatJ version 3.04.6[25] and details are provided in Supplementary Table 2. Gray shading indicates the interquartile ranges (IQR, the 75th and 25th percentiles). Units are Tg P yr$^{-1}$. Note that the Soviet Union is assigned as part of East Europe before its collapse and countries are assigned into different continents according to their geo-location after the collapse. Source data are provided as a Source Data file.

Thailand 2%. Reduction in P-harvest was the main contributor to reduction in P-net from a few countries such as Japan, Russia, Chile, Denmark, and Canada.

We then address the question of scenarios for the future evolution of P-net and how the fish production sector could become P-neutral by year 2050. Wild fish production has stagnated for the past two decades and is unlikely to grow significantly by 2050; on the other hand, the "business as usual" aquaculture production is projected to reach 140 Tg (or 2.3 times its 2010 level) in 2050 after accounting for fish supply and demand, health of wild fisheries, fish prices, population growth, GDP growth, and technological progress[26,41]. We first establish a *baseline* scenario to 2050 by assuming that wild fish capture will be stable at its mean level of 2005–2014. We calculate that P-input to aquaculture will grow up to 3.42 Tg P yr$^{-1}$ to match the fish demand increasing from the "business as usual" aquaculture production projection from Waite et al.[26] with PUE staying at its current level of 20% (see Methods). In this *baseline* scenario, P-net will remain negative, and the imbalance between land and aquatic P flows will grow by a factor of two to reach −2 Tg P yr$^{-1}$ by 2050 (Table 1). We then build an idealized alternative scenario where a neutral P-net is set as a global target by 2050. To meet this target, the weighted (by fish production) global average PUE will have to increase to ~48%. To assess if such a PUE increase from current low value to a high efficiency of 48% could be achieved with current best practices, we analyze the upper range of PUE among diverse aquaculture production systems, and upper range of PRE data from feeding experiments. The rationale is that PUE could be increased by transforming practices in all aquaculture farms so that they could reach their currently maximum achievable value.

The PUE of individual farming system spans a wide range (1–167%, Supplementary Fig. 5). Our PUE database covers six of the seven largest aquaculture P-input contributors, including China (59% of P-input), India (10%), Vietnam (6%), Bangladesh (3%), and Thailand (2%) but excluding Indonesia (5%). Thus, the global weighted average PUE (20% during 2005–2014) mostly reflects the PUE from China due to its dominant share. In China, the current median PUE is 19% from finfish and 12% from

crustacean species, and the upper values are 44% (95th percentile) for finfish and 24% for crustacean species. Adopting co-culture farming systems (finfish + crustacean) may also increase PUE, by up to 61% according to one study[42], but there is a large uncertainty in the PUE of those systems, with few data available[43,44] (Supplementary Discussion: Data pattern).

Upper values of PRE measured in controlled experiments are also informative: they indicate a potential gain in PUE that could be achieved in (idealized) closed culture systems with no leakage. The overall median PRE is 37% based on 348 feeding experiments conducted in controlled environments (e.g., closed tanks) (Supplementary Fig. 6), that is, 1.85 times larger than median PUE. The 75th percentile of PRE is 52% from finfish and 21% from crustacean species, respectively. The 95th percentile reaches 78% for finfish and 26% for crustacean species. Finfish dominated the farmed fish production and contributed to 87% of total P in harvested fish (mean, 1950–2016), while the share of crustacean species grew from near zero in 1950 to 6% in 2016 due to the decline in cultivating mollusks (Supplementary Fig. 7). If we assume a PRE of 21% from raising mollusks and with 87% of harvested fish P from finfish, the 75th percentile of PREs would correspond to an aquaculture PUE of 48%. The most important cultivated fish species, the carps, comprise ~40% of the total aquaculture production by weight[45] and their 75th percentile PRE reaches 57%.

**Implications for the large-scale P budget.** Human perturbation of the global P cycle has mobilized a large amount of P from phosphate rock into the hydrosphere. The total P load into aquatic ecosystems from crop livestock spans a large range among studies and is reported to be 4 Tg P yr$^{-1}$ (fertilizer and manure sources only) in 2000 from Bouwman et al.[46], 5 Tg P yr$^{-1}$ (fertilizer and manure), or 13.5–25 Tg P yr$^{-1}$ if land use change was additionally accounted from Peñuelas et al.[7] over 2005–2011, 12.9 Tg P from Chen and Graedel[47] in 2013, and 9.7 Tg P yr$^{-1}$ from Lun et al.[6] during 2002–2010. The global fishery P input into aquatic ecosystems reached 2.06 Tg P yr$^{-1}$ in 2016, which is significant despite smaller than P load through crop livestock.

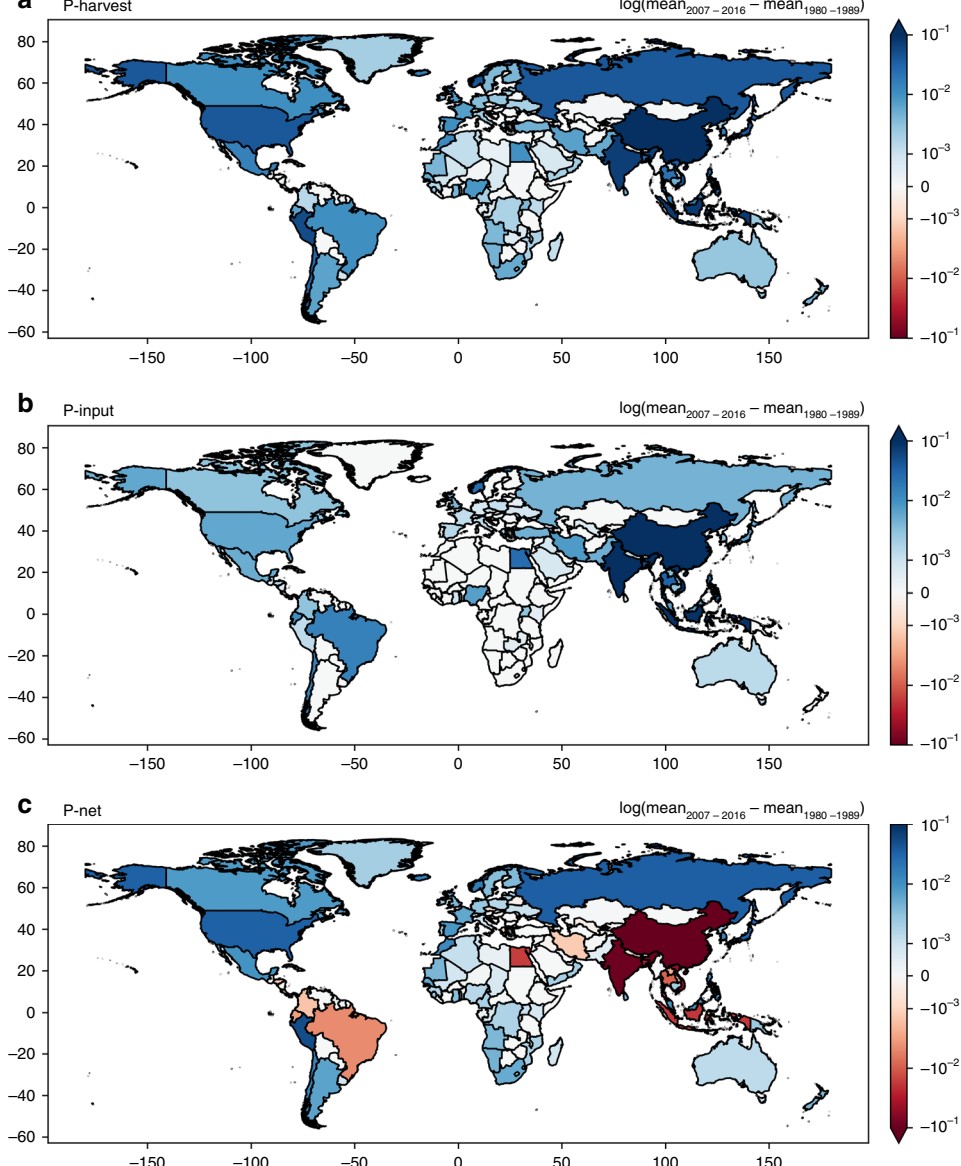

**Fig. 4 Changes in the global fishery P budget.** (**a**) P-harvest, (**b**) P-input, and (**c**) P-net between the 1980s and the recent decade (2007–2016). Note the log scale on the color legend and that the units are in Tg P yr$^{-1}$. Country borders are adapted from TM World Borders Dataset 0.3.

### Table 1 Projections of mean global fishery P budget in 2050.

|  | Wild production | Aquaculture production | P-harvest | P-input | P-net |
|---|---|---|---|---|---|
| Unit | Tg | Tg | Tg P | Tg P | Tg P |
| Mass | 98.7 | 140.0 | 1.41 | 3.42 | −2.01 |

Per unit food protein supply, P-net from fish-driven food production (capture fisheries and aquaculture) ranges from 0.019 (here positive means a landward P movement) to −0.052 g (P) per g (Protein), which is still smaller in magnitude than that from the crop–livestock system, −0.025 to −0.138 g (P) per g (Protein) (Supplementary Discussion, Supplementary Table 1). If historical fish P protein had been supplied by the crop–livestock system instead, there would have been more P loaded into aquatic ecosystems. However, when we investigated aquaculture separately, P-net went from −0.292 to −0.310 g (P) per g (Protein) (Supplementary Table 1). Some P from crop livestock is used as

food and fertilizer for aquaculture. The global fishery P-input that is not accounted for in the crop–livestock P loads, such as through compound fish feeds and mineral fertilizers, is nontrivial. For example, the global total of commercially manufactured compound aquafeed is estimated to be 34.4–39.6 Tg in 2012[48]. An average of 1% P concentration in compound aquafeed corresponds to more than 0.3 Tg P yr$^{-1}$ input into aquatic ecosystems. In addition, crop-livestock P that enters aquatic ecosystems relies on complex interactions between local topography, vegetation cover, climate, and land–water connections that regulate the fraction of surplus P going through erosion, runoff, and leaching[37]. Fish feeds and P fertilizers are commonly directly dumped into receiving waters to support fish production, resulting in the fraction of P that ends up in the water (commonly used as a parameter to estimate P loads)[37] being different from the crop–livestock system, especially for marine culture in coastal regions. Leaking P from aquaculture system that ends up in the ocean may contribute to feed wild fish. It is likely to be marginal considering the large amount of P available in the ocean

at a global scale but might be important locally. We acknowledge that not all P makes its way to the ocean. For example, in some ponds, up to half of the nutrients may end up in sediments that can be reused for agriculture[49]. In all, 84–94% of the aquaculture P-input went into the freshwater and only a portion could ultimately enter the ocean. Such internal P loading, if not managed properly, would result in the legacy P problem that might extend poor water quality issues for decades after adopting good management practice[50]. The higher aquaculture P-input from our data-driven estimation compared with the modeling result of Bouwman et al. [46] and Bouwman et al.[51] can partly be explained by the fate of P after entering the aquatic ecosystems. For example, Bouwman et al.[46] quantified P release from ponds and assumed that particulate P was not released from pond systems, while our study quantified the total P-input into aquatic systems that include ponds.

Through a data-based global quantification of the underrepresented yet critical land–aquatic P fluxes, we found the largest pathway that transferred P from aquatic ecosystems to land, which has important implications for the biogeochemical cycle of P. In the 1950s, total global P fertilizer application amounted to around 3 Tg P yr$^{-1}$, of which 1 Tg P yr$^{-1}$ ended up in aquatic systems[52]. A net P transfer of 0.2 Tg P yr$^{-1}$ in the 1950s from the global fishery is a relative significant contribution to return P from aquatic systems to land. Historically, human demand for fish may indirectly alleviate P shortage on land as wastes associated with fish processing or consumption would serve as crop fertilizer or feeds for livestock. Compared with current known pathways that transfer P from aquatic to land systems (0.0056–0.16[34–36] Tg P yr$^{-1}$), fish-driven landward P flux was the biggest yet largely overlooked, at least in the 1980s. We also revealed a shift of the net land–aquatic P transfers driven by human demand for fish. If no waste management practice is implemented to reuse P accumulated in aquaculture systems, a net global fishery flux of −0.95 Tg P in 2016 (or −2.01 Tg P in 2050) is considerable. Regionally, the threat of crossing the planetary boundary is even stronger for Asian countries, and this is without considering fish trading that may alter regional P fluxes. Future studies focus on the spatial heterogeneity of fishery P fluxes, the costs, and who experiences that the impacts would be helpful in clarifying environmental and ecological benefits or harms locally.

**Outlook for a global P-neutral fishery management.** Fish have a higher feed conversion efficiency (e.g., compared with beef and pork) and fish production's carbon footprint is lower than other animal production sectors[22]. As a result, aquaculture production is expected to grow further to sustain future demand for food[22]; it follows that P-input is going to increase unless mitigation strategies are implemented to improve aquaculture PUE. We could not detect an improving trend in PUE in our dataset because of the lack of repeated measurements over time. Nevertheless, we are optimistic in aquaculture becoming more efficient with modern technology and management playing a more important role in future.

Technically, the difference between median versus upper PUE values (e.g., 61% from China) and optimal PRE values (52% or 78% from finfish) suggests that it should be feasible with current technology to increase the current global average PUE (~20%) up to the 48% target in 2050. This increase might be achieved through optimizing feeding efficiency (e.g., modifying formulated diets, feeding frequency, and culture environment) and promoting low-impact production systems (e.g., recirculating aquaculture systems, biofloc technology, and integrated systems). Integrated systems, such as the integrated multitrophic aquaculture (IMTA) and integrated aquaculture/agriculture (IAA), use wastes from one species or one subsystem (e.g., aquaculture, crop, and livestock) to serve as food or fertilizer for another. These systems incorporate ecological principles to recycle nutrients, and in theory, are self-sustainable. The low PUE from crustacean species could potentially be improved through dietary phytase supplementation or coculture with finfish species[42–44].

In practice, despite researches documenting the benefits of integrated rearing systems, a large fraction of nutrients is unused and they are not recycled[31], as witnessed by the decline of traditional IAA, especially in China where there are increasing concerns about environmental sustainability[53]. Modern IAA, or modern ecological aquaculture[54], is promising: it improves aquaculture PUE but requires effective management and modern technology to upgrade traditional semi-intensive practices. Ref. [53] reviewed possible ways of successfully combining traditional ecological aquaculture with modern pellet-fed methods. With modern ecological aquaculture, a PUE of 48% would become an achievable target technologically, despite economically there is a long way to go. Improving PUE in aquaculture is urgent in China[55] and China is playing a leading role in restricting aquaculture P-input by adopting strict regulations aiming for a green growth of its aquaculture during the recent decade[56]. It is possible to expect a raise in aquaculture PUE in China in the future with its aquaculture practice gradually shifting toward environmental sustainability[57]. Countries like India, Vietnam, Bangladesh, and Thailand, which contribute significantly to aquaculture, yet with a lower PUE than China, need more support in adopting advanced feeding technology and modern practice.

We need to pay attention to the role of capture fisheries and aquaculture in returning P back into the land–human systems. A wide spectrum of technologies exist that would allow P to be recovered from heterogeneous waste flows, including fish-processing waste, transportation losses, residential, commercial, and institutional food waste, human excreta, sewage networks, and landfills[58]. For example, treated fish-processing wastes are widely applied as animal feed and soil fertilizer[59–61]. In total, 70% of fish is processed before delivering to the customer, resulting in 20–80% of fish waste depending on final products and fish species[60,62]. China's fish-processing industry could produce 0.65 Tg of fishmeal and 0.16 Tg of fish oil[60]. Globally, 10% of fish P harvest was returned to aquaculture through fishmeal and fish oil, and another 4% was recycled as fishmeal and fish oil for poultry, swine, human supplements, and other sectors, leaving 0.84 Tg P yr$^{-1}$ of harvested fish P that could potentially be further recovered and reused (see Supplementary Discussion for details) in 2010. The proportion of fishmeal and fish oil used to feed farmed fish is decreasing, partly due to the temporal instability in supply, improved feeding technology, and rising prices, and this trend is likely to continue in the future[63]. That means more harvested P can potentially be reused for enhancing other aspects of the human food chain in the future. Recovering P from urine and feces has a long history and is being advanced by numerous studies today[64–67]. Recovering P from wastewater has been a priority subject of research with a range of scientific and engineering advances being made[68,69]. Many other strategies focusing on food and waste management are also being put into practice, e.g., the diversion of food waste from landfill to agricultural land by composting[70] and the application of decentralized P recovery systems[71]. These technologies are promising to recover most P transferred from the aquatic to land–human systems through harvested fish in future. As an example, Norway is at the top of human development index rankings and has a P use efficiency of 92% in fishery with high

utilization of fish scraps from land-based processes[72]. P reuse and recovery are currently far from mainstream practice[58]. Practical issues such as costs and mismatches between locations may discourage P recovery practice. We need sustainable policies at the international, national, and local levels to develop efficient P recycling strategies and sound management plans to motivate industrial and household-level practice. Although fish trading complicates country-level commitments, countries with high P-harvest, from China, Indonesia, India, Peru, Norway, Russia, United States, and Japan to Vietnam, have a crucial role to play in leading the practice of recovery and reuse of harvested fish P.

The global fishery does not occur in isolation—the fishery P dynamics are closely coupled with other P mass flows and is one part of the global P biogeochemical cycle. Cultivated fish species increasingly depend on feed inputs from the crop–livestock system[30,73], while wild capture fishmeal and fish oil are traditionally used to feed livestock. Researches focusing on food, biodiversity, climate change[74], and land use[27] have now started to integrate wild capture fishery, aquaculture, and agriculture in assessing sustainability challenges. We propose that future studies should incorporate fishery into P assessment and mitigation strategies. As the case study in Norway[72] shows, P use from both within and cross-sectors of aquaculture, fisheries, and agriculture is far from being optimized. We need integrated management systems that simultaneously optimize PUE from multiple sectors. A global effort to optimize, integrate, and manage multi-sectorial P dynamics is the path to a sustainable blue revolution in aquatic systems.

## Methods

**Overview**. Our ability to quantify large-scale fishery P is restricted by the limited amount of data available. We compile large datasets on the whole-body P concentration of different fish species, PUE at the culture-system level, and PRE from controlled feeding experiments (see Methods: Data and Supplementary Discussion: Data pattern for details). Supplementary Fig. 15 is a schematic diagram showing the global fishery P budget calculation. Harvested fish P mass is directly calculated from fish live biomass and P concentration (per unit live biomass) for different species (Methods: P budget). Total aquaculture P-input can be estimated through total aquaculture area, and feed and fertilizer inputs per unit area[28], or through fish biomass production and the ratio of harvested to input P (phosphorus-use efficiency, PUE). We apply the latter approach due to the lack of a direct survey of global aquaculture area, and feed and fertilizer P input per unit area. Ref. [52] modeled finfish aquaculture P-input of major finfish species based on harvested fish P mass, feed conversion ratio (FCR, the ratio of feed biomass to fish biomass), and P fraction in feed. Here we use a different approach to estimate aquaculture P-input based on harvested fish P mass and PUE at culture-system level (Methods: P budget). We do this, first because aquaculture is highly heterogeneous. Culture-system-level PUE takes into account diverse aquaculture practices in the real world, which might deviate from PUE or P retention from idealized feeding experiments focusing mostly on a single species (Figs S5 and S6). Second, the PUE at culture-system-level approach avoids the use of additional variables (such as P fertilizer input rate and P fraction in different feeds) for which insufficient data are available. We further separate the global budgets by continents and countries, and project into the future. The PRE database is applied as a reference to assess whether a P-neutral fishery is technically achievable in the future.

**Data**. World fishery production (1950–2016) is obtained by combining two databases, the Food and Agriculture Organization of the United Nations (FAO) Global Fishery Production database, FishStatJ version 3.04.6[25] and the reconstructed wild marine fish capture database from Sea Around Us (http://www.seaaroundus.org/).

FishStatJ 3.04.6 tracks the annual nominal capture of wild finfish, crustaceans and mollusks, the harvest of aquatic plants and mammals, as well as fish harvested from aquaculture and other fish farming from fresh, brackish, and marine waters. The database separates production from different countries, for multiple species, and across diverse water areas around the globe from 1950 to 2016. In line with the FAO annotation, fish in this study refers to finfishes, crustaceans, mollusks, and other aquatic invertebrates. Total fishery production increased from 21 Tg in 1950 to 205 Tg (including aquatic plants and mammals) in 2016[25], while finfish, crustaceans, and mollusk production increased from 19 Tg in 1950 to 169 Tg in 2016[25]. Aquatic plants, overwhelmingly seaweeds, can remove nutrients from the aquatic ecosystems because farming of aquatic plants does not require large supplementary P input. However, with a relatively low P concentration, aquatic

plants' contribution to the fishery P budget is unlikely to be substantial. Ref. [51] showed that seaweed removed <0.01 Tg P yr$^{-1}$ from aquatic ecosystems. In this study, we do not include aquatic plants. Aquatic mammal production ranges from 0.19 to 1.11 Tg, which is also not included in our P budget.

FAO acknowledges its concerns on the quality of biomass production data, and data reported by FAO on behalf of reporting countries are incomplete with under-representation of nonindustrial (e.g., artisanal, subsistence, and recreational) and illegal fisheries[75–77]. Therefore, we use the reconstruction of wild marine catches from Sea Around Us to correct the FAO database for the underrepresented fish harvest. This decade-long catch reconstruction integrates collective efforts from hundreds of experts and covers the whole globe. Ref. [78] showed that the reconstruction from Sea Around Us is on average 53% higher than that given by earlier FAO reports (1950–2010), including fish that are not landed, i.e., discarded fish. Discards refer to fish that are not retained on board during fishing operations and are returned to water bodies. As discarded fish do not reach land, we assume that discarded fish do not play a role in land–aquatic P transfers. FishStatJ 3.04.6 wild fish production is scaled by landed fish from Sea Around Us at the country level from 1950 to 2014 with the average scaling factor decreasing from 1.29 (1950) to 1.06 (2004). The scaling factor for year 2014 is applied to year 2015 and 2016 as Sea Around Us only reported data up to 2014. The problem of misaccounting of fish biomass is not significant in aquaculture, at least mariculture[79], and we use the FAO dataset for aquaculture. Production is expressed as per unit live weight (wet weight).

P is unevenly distributed in fish organs. The skeleton (bone or cartilage) generally has a higher P concentration compared with muscles. Existing food nutrition databases, such as the FAO/INFOODS global food composition database for fish and shellfish—version 1.0 (uFiSh1.0), report P concentration of edible portions of fish, but this does not adequately capture the whole-body P concentration. We compile data from the available literature that report whole-body P concentration of finfish, crustacean, and mollusk species. A total of 175 peer-reviewed studies were compiled in the database (Supplementary Data 1). The entire database includes 262 records for wild and 902 records for raised fish across diverse species (224 in total), environmental conditions, dietary treatments, and ontogenetic stages. Among these records, 1088 entries document P concentration for finfish, 41 for crustaceans, and 35 for mollusks.

The database reports whole-body P concentration in mass fraction, i.e., the mass of P divided by the live mass of fish. When a specific study reported dry weight-based P concentration, we transfer P concentration into wet basis using fish moisture content. If the study does not report fish moisture content, the average moisture content across the entire database is applied for this unit conversion. Fish name and auxiliary information such as living habitat, if not reported, are obtained from FishBase (http://www.fishbase.org). Supplementary Figs. 3 and 4 display the distributions of whole-body P concentrations.

PUE is defined as the proportion of supplementary P applied to feed fish and fertilize the aquatic ecosystems that are recovered in harvested fish. Culture-system-level PUE quantifies the actual P investment considering various environmental and practical factors. The PUE database contains 168 cases from 96 peer-reviewed publications that directly track culture-system-level P budget, including, at least, P harvested through fish and P input through feeds or/and fertilizers (Supplementary Data 2). Culture systems cover pond, tank, cages, recirculating, and flow-through aquaculture systems across India, United States, Madagascar, Thailand, Mexico, Brazil, Ireland, China, Honduras, Czech Republic, Sweden, Bangladesh, Israel, Australia, France, Vietnam, Poland, and Saudi Arabia. Feeding experiments frequently quantify the ratio of P that is incorporated into fish biomass to the total fish P intake (PRE, Supplementary Data 3) at a single fish-species level. We do not incorporate feeding experiments into our PUE database if the study does not track the P budget for the whole culture system. Species-level PUE is close to culture-system-level PUE in monoculture. However, polyculture systems are a common practice where culture-system-level PUE might deviate from species-level PUEs. For example, four of the most widely raised species, silver carp (a photoplankton filter feeder), grass carp (a herbivorous macrophyte feeder), common carp (an omnivorous detritus feeder), and bighead carp (a zooplankton filter feeder) are frequently cultured together. Environmental conditions affect the portion of feed that is not taken up by fish, and waste from one species can serve as food for another in polyculture. The P pre-existing in water used to raise fish is not considered as an external P input because we focus on land–aquatic P exchanges (Supplementary Methods). Supplementary Fig. 5 shows the distribution of culture-system-level PUE.

**P budget**. Annual P-harvest is estimated from Eq. (1) by summing P mass from each harvested species.

$$P_{\text{hat}} = \sum_{i=1}^{n} W_i * R_i,$$  (1)

where $P_{\text{hat}}$ (Tg P yr$^{-1}$) is the annual P-harvest, $W_i$ (Tg yr$^{-1}$) is the weight of fish live biomass production, and $R_i$ is whole-body P concentration (P mass fraction per live biomass) for each species $i$. The mean whole-body P concentration is used for each fish species, averaging over a variety of living conditions, ontogenetic variations, and physiological status. For species that have no corresponding whole-body P concentration compiled in the whole-body P database, the mean P concentration

from the same taxonomic order was substituted as a proxy. The whole-body P database covers >80% of fish production at the taxonomic rank of order or lower. For the remaining 20%, the major group level (finfish, crustacean, and mollusk) mean whole-body P is applied. For example, cartilage fish (e.g., sharks, rays, and chimeras), which may be structurally different from bone fish, comprise around 1% of the total captured fish biomass; they are not included in our whole-body P database and the average finfish P concentration is used as an estimate. The class "other aquatic invertebrates" covers fish that do not belong to any of the three major groups reported on by FAO; they account for less than 0.7% of total fish biomass and are assumed to have a mean P concentration that is the same as that of the entire database.

Uncertainty in harvested P estimation caused by variations in whole-body P concentration and fish biomass estimation is assessed by the Monte Carlo method. We randomly sample the whole-body P concentration 1000 times for each species in our database and calculate P-harvest budget. The number of compiled whole-body P concentration entries differs for different species. P concentration for each species may not cover the whole range of uncertainty. To avoid under-estimation of uncertainty associated with P concentration, we also make use of information from higher taxonomic levels. If the number of P concentration entries are no more than a cutoff criterion (one record as a start), then the random samples for the corresponding species are generated from the selection pool with P concentration information covered by the taxonomic order that this species belongs to. P budgets are not sensitive to the cutoff criterion that determines the population from which random samples are drawn (see Supplementary Discussion: Uncertainty and sensitivity); we therefore report results from setting the cutoff criterion to be one entry. When the species-level average is different to the average of the taxonomic order level, random samples taken directly from the bigger taxonomic order pool would deviate the average P concentration from the species-level pool. In order to keep the same average while covering a reasonable uncertainty range, we conduct normal sampling using the species-level average and the standard deviation that is rescaled by the coefficient of variation at the taxonomic order level. Similarly, when the taxonomic order level P entry is no more than the cutoff criterion, the scope of the random sampling lies in the major group. Here, we prefer to use mean, instead of median, to represent the average P concentration, as the median might be biased by the number of Monte Carlo samples when the population size is small. The random sampling strategy is evaluated by changing the criterion based on which information is extracted from the order or major group level (Supplementary Discussion: Uncertainty and sensitivity; Supplementary Fig. 9).

Uncertainty or accuracy of fish biomass production data is debatable, and is not straightforward to assess[75–79]. One alternative strategy is to assign scores, which are associated with certain categorized uncertainty levels, for each data source that contributes to fish biomass production based on expert judgments[78]. We do not trace back the uncertainty to each data source as it is hard to establish a common standard for the scoring system from different countries and based on a variety of reporting sources. Instead, we look into the uncertainty by assuming a range of uncertainty levels (0%, 50%, and 100%) to assess the contribution of biomass uncertainty to the overall uncertainty estimation of P budgets (Supplementary Figs 9 and 10). A 50% percentage uncertainty corresponds to very low confidence, that is (quoting the IPCC[80]) "less than high agreement and less than robust evidence". As a conservative strategy, we report results at the 50% uncertainty interval in the main text and the sensitivity of P budgets to fish biomass uncertainty is provided in Supplementary Discussion: Uncertainty and sensitivity. We randomly sample fish biomass within the corresponding percentile uncertainty and conduct 1000 Monte Carlo calculations for each uncertainty level.

A wide range of feeds and fertilizers, varying in ingredients and weight, is applied in aquaculture[81]. Fish feed includes commercially manufactured compound aquafeeds with different additives, farm-made feeds from crop–livestock products and/or by-products, such as animal liver, blood, meat, offal, poultry feathers, bone, and eggshell, but also barley, cereals, maize, rye, sorghum, wheat, soybean, oilseed, coconut, cottonseed, and terrestrial invertebrates (silkworm pupae, maggots, soldier flies, locusts, termites, and earthworms), and fish with low economic value or by-products from aquaculture industry[81]. Species, such as silver and bighead carps, filter-feeding fish species (e.g., bivalve mollusks) require no external feeds and are not fed. Instead, P fertilizers are applied to increase aquatic primary productivity to support those species that do not rely on external feeds. Fertilizers cover inorganic chemical fertilizers and manures from a variety of sources, such as poultry, ruminant waste, swine waste, and human excreta[81].

Drawing up a direct bottom-up estimate of feed and fertilizer P investment is difficult due to scarce data. Globally, total commercially manufactured compound aquafeed is estimated to be 34.4–39.6 Tg in 2012[48]. However, the global estimates of feed from farm-made products and low-value fish are largely undocumented. Farm-made aquafeeds are roughly reported to be between 18.7 and 30.7 Tg in 2006[82], and low-value fish is estimated to be 5.6–8.8 Tg. Insufficient data are available to quantify the global total of mineral fertilizer applied to aquaculture globally. The volume of manure that goes into aquaculture and its phosphorus concentration are also largely unrecorded.

Therefore, we estimate external aquaculture P input through culture-system-level PUE (per feed and fertilizer applied). Due to the large variation and limited

data coverage, we cannot detect significant differences ($p > 0.1$, Student's $t$ test) in PUE between rearing systems (e.g., pond or cage), between countries, or between freshwater versus marine aquaculture, with the exception that finfish farming has a significantly ($p < 0.01$, Student's $t$ test) higher PUE compared with the farming of crustacean species. Advancements in feeding technologies can improve feed conversion ratio (FCR, the ratio of feed biomass to fish biomass) and thereby reduce feed requirement; however, in practice, its impact on PUE depends on economic, policy, and societal drivers such as the farmers' socioeconomic status, attitude, and behavior. Our PUE database does not support the occurrence of a statistically significant shift in PUE through time. Because fish production databases report fish live weight separately for fresh and marine (including brackish water) environments, we also differentiate PUE between fresh and marine systems and among finfish, crustacean, and mollusk species. The culture-system-level PUE database is separated into six groups: freshwater finfish, marine water finfish, freshwater crustacean, marine water crustacean, freshwater mollusk, and marine water mollusk. For each group $j$, total feed and fertilizer P input is estimated by dividing harvested P mass ($P_{hat,j}$) by culture-system-level PUE ($PUE_j$, Eq. 2). Global feed and fertilizer P input is the sum of P inputs across these six groups. The distribution of PUE deviates from a normal distribution (Supplementary Fig. 5) and the median provides a better representation of typical PUE. Currently, PUE of marine water mollusk is used for freshwater mollusk as we do not have experimental studies on PUE of this latter category in our database

$$P_{inp} = \sum_{j=1}^{6} P_{hat,j}/PUE_j, \qquad (2)$$

where $P_{inp}$ is the P-input. Factors that may affect PUE were not studied individually but are taken into consideration in the uncertainty estimation. We conduct 1000 Monte Carlo calculations randomly sampling PUE (within each group) through the compiled PUE database that incorporates a range of variation (see also Supplementary Discussion: Uncertainty and sensitivity). The final P budget uncertainties are characterized by the interquartile ranges (IQR), because the distribution of PUE is highly skewed, and broader ranges are more likely to be affected by extreme values partly associated with the limited sample size of our database.

**Reporting summary**. Further information on research design is available in the Nature Research Reporting Summary linked to this article.

## Data availability

Databases of fish P concentration, culture-system-level P use efficiency, and P retention efficiency are provided as Supplementary Data. Additional data that support the findings of this study are available from the corresponding author upon request to the corresponding author. The source data underlying Figs. 2 and 3 are provided as a Source Data file.

## Code availability

Calculations were conducted through Python 2.7.15 and the code is available upon request to the corresponding author.

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

## Acknowledgements

This work was financially supported by the IMBALANCE-P project of the European Research Council (ERC-654 2013-SyG-610028).

## Author contributions

Y.H., P.C., D.G., J.S. and J.P. designed this study. Y.H. and J.S. collected the data. Y.H. conducted the analysis and drafted the paper. Y.H., P.C., D.G., J.S., J.P., F.C. and H.Z. discussed the results and contributed to the paper.

## Competing interests

The authors declare no competing interests.
