## [Peer Review File · Nature Communications]

Reviewers' comments:

Reviewer #1 (Remarks to the Author):

This paper nicely shows how the transfer of phosphorus (P) via fisheries has shifted on a global scale over the past several decades. The findings are quite interesting and certainly relevant to global environmental issues, food security and sustainability. Overall the methods seem robust and the paper is well written. I think this will ultimately be an impactful publication, although I do have several suggestions that I think will improve the paper.

One general issue, which I think is common for 'budget' studies such as these, is that it is important that the authors clearly define the spatial scale, i.e., the boundaries and fluxes. For example, the study includes fluxes to and from both marine and freshwater systems, but in several places the authors seem to be most concerned with the oceans, i.e., how much P is delivered to vs. removed from the oceans. I understand that the oceans can be considered the 'end point' of P flows, but from an environmental perspective, net retention of P in freshwaters is just as important as that in marine systems. Furthermore, the emphasis on marine systems could confuse readers, who might think that freshwaters are included in the 'land-human' system (as is sometimes the case for global carbon budgets, for example). Therefore, I ask that the authors 1) clarify that 'aquatic ecosystem' includes both marine and freshwater (e.g., early in the text and in the legend for Fig. 1), and 2) omit statements that could cause confusion. The text on lines 260-263 is a good example of this – why single out ponds? They are aquatic ecosystems too, so in principle human-derived P that is stored in ponds is the same as human-derived P stored in the oceans (in terms of importance, and certainly in terms of the budget).

The authors mention that reliance on 'fish' (including shellfish) is likely to increase because of increasing demand for protein. I wonder if they could estimate the relative consequences of reliance on aquatic protein vs terrestrial protein. For example, if the increase in fish harvest since 1950 was replaced by an increase in terrestrial protein, what would happen to the P budget? I realize this is a bit beyond the scope of the paper but nevertheless some quick calculations might be illuminating and would be of great interest.

Line 95: insert 'to aquatic ecosystems' after 'add' to make it clear what the fluxes are.

Line 97: is 'production' the same as harvest but just different units? (mass for production, P for harvest)?

Line 160: Seems like Russia/Soviet Union should also be included here.

Figure 1 is really nice, and I wouldn't be surprised to see this reproduced in textbooks in the future.

Figure 4: The units here seem unnecessarily confusing. Just to be clear, a value (in the legend) of 0.01 would mean 1 Tg P yr⁻¹? Why not just relabel scale to Tg?

Lines 244-247. This is a very confusing sentence. As mentioned earlier, with budget studies like this, it's really important to keep things clear, and I think this will confuse many readers.

Lines 286-287. It is curious that efficiency is higher for fish than invertebrates. Fish have much higher body P concentrations than invertebrates. Thus, invertebrates are likely to consume food that has a similar P content to their own bodies, which would seem to render them more efficient than fish at converting food P into body P. Is there a logical explanation for the lower efficiency of invertebrates?

Line 292 and nearby: I don't fully understand where this number (0.84 Tg P yr⁻¹) comes from.

Line 343: Did calculations use a constant PUE among countries (within a particular culture-system), or was PUE country-(region)-specific?

Paragraph beginning on line 444. I don't really understand how uncertainty in body P was explored. What is meant by '1 record as the default'? I think this has to do with sample size for body P content for specific taxa, but it's very hard to follow, even after reading the Supplemental Information.

Data: I have a few questions regarding data on body P. First, the authors refer to these data several times, but I cannot find the data. It is not in any tables, and I don't see any data files accompanying the manuscript submission. I can't even find any reference to the data base. It would be nice to have access to these data. Also, what do sample sizes refer to in Fig. S3? The number of studies? The number of individuals? Either way, the sample sizes seem too low, especially for the two invertebrate groups. But this depends on the scope of the analysis. For example, papers by Ikeda contain several hundred measures of body P contents for crustaceans, but most of these are for zooplankton such as copepods. Were these values excluded because zooplankton are not harvested? The authors refer to a 'global' database, but it's not clear if this means global only in a geographic

sense, or also in terms of the taxonomic pool being considered. In the end, I doubt that this will affect the conclusions because uncertainty in body P content seems to have little effect on P harvest rates at the global scale. Nevertheless, it would be best to clarify these points, and to include the data.

Supplement, Section 3 (Harvested P to be used by land activities). I understand what this means, but just to be clear, the basic budget (and P-net in particular) did not take this into account, correct?

Fig S7b: This is very confusing with the different scales and colors. Also the legend doesn't make sense to me.

Supplement, Section 7 (Recycling and reuse of harvested P). This section seems like 'Discussion' and therefore seems out of place in the Supplement.

Reviewer #2 (Remarks to the Author):

This paper provides a global review of phosphorous cycling associated with fish and seafood. From the 1950s-1980s, the seafood system was dominated by wild capture, and shifted P from aquatic to terrestrial systems. More recently, feed-driven aquaculture mean the seafood system brings terrestrial P, in the form of fertilizers, to the aquatic systems. Seafood thus represents a net deposition of .96 Tg P yr⁻¹ into aquatic systems, a meaningful portion of net deposition from all sources.

I am reviewing these results from the perspective of someone knowledgeable about global fish and aquaculture policy discussions, where they aim to have an impact; I'm not

a biogeochemist or life cycle analyst. With this lens, I appreciate the scale of the calculation exercise here, but the chosen scope and scale of the analysis does not seem to provide the most important insights. Rather, I am left concerned that it will mislead consumers or other users of this information towards other protein sources, which are in fact worse than the environment.

Major Comments

There are three places where I struggled to understand the importance of this work. First, it is not clear from this article why transfers of P to aquatic systems are a problem. What are the costs of this, and who experiences them? The planetary boundary is referenced, but is the issue really planetary, or is it local? Lines 92-94 say the final disposition of P here isn't the focus of the study, but then why does the study matter? Where is it OK for P to end up?

Second, it is odd to me to scope the study to combine capture fisheries and aquaculture, as if there is a natural balance in the two processes simply because they both result in fish in the market. Why does capture fisheries' net terrestrial shift only offset aquaculture's net aquatic shift, rather than the net aquatic shift driven by intensive food production for all proteins? What if P contributions could be more effectively reduced through other proteins? Rather, the importance to highlight seems to be shifting from capturing wild fish which use natural ambient P, to application of chemical fertilizers, which are much different processes. Each of these sources might be of interest as components of the food system level, but it is not clear that I've learned something about the state of the world by aggregating the two, to the exclusion of other methods of food production.

Third, such environmental impact studies are frequently taken in the fish literature to scare consumers away from fish, depicting it as environmentally unfriendly. However, this is only good for the environment if what consumers would eat instead has a lower P footprint. Thus, some additional context is necessary to compare numbers reported here with what is known about P from other protein sources, or at least to be circumspect about the fact that reducing P from aquaculture might lead to more P someplace else in the food system.

If there's all this runoff feeding aquatic and marine systems, isn't it helpful to extract it (by catching fish)?

Minor Comments

Does fish mean only finfish, or also invertebrates?

32: having trouble reconciling at "safe global" level with preceding discussion of observed local saturation. What does a global number mean?

54: This is a very broad statement to make about aquaculture in particular, separate from agriculture.

61: What is considered state-of-the-art?

89: This is feed and fertilizers specific to producing fish, yes?

Fig 1: Fish aren't fertilized P directly, right? This goes to plant-based feed components. Also, harvested fish should link to fish feed, through meal. This diagram should be much clearer about the transport pathways.

119: "P-harvest" is P from capture fisheries only?

123: Again, this is just farming for aquaculture feed? Needs to be clearer what this is.

124-128: This is hard to unpack.

137: This might be one of comparisons I asked about above, but I cannot tell exactly what is being said here. This is runoff losses?

141-43: I cannot tell what this says.

189: This feels like a "scare" number that may make little sense practically. "Business as usual" sounds like the bad IPCC scenario, but that might not apply here. In particular, if people don't eat aquacultured fish, what will they eat, and will that be better for P? Also, what are the incentives for feed producers to increase PUE? Is there a reason to think it would be unchanged? If this can't be done sensibly, it should be removed from the paper.

196: This assumes variation in PUE driven by heterogeneity in the efficiency of using a single technology, rather than heterogeneity in technologies used to produce various species in different environments. Does this really provide a good indication of potential PUE?

244: Not clear exactly what is being counted here, especially with such a huge range, and probably considerable reference year dependence.

246: How does this lead to double counting? Does this arise from where things are measured, so they might be utilized "downstream" of the initial point of measurement?

252: Is this gross or net of utilization?

275: Again unclear on what the planetary boundary means in this application. What will this look like?

281: This is an odd goal. Why credit aquaculture with the benefits from capture fisheries? Shouldn't the goal be to reduce pollution from each form of production to a level where the benefits of marginal abatement equal the marginal costs?

283: It is not obvious to me why net landward is good. Alternatively, how much would global capture have to increase?

289: But total P-input might not, if people are eating fish instead of more P-intensive proteins.

301: Is this true? Many least developed countries rely more on capture than aquaculture.

310: Not necessary?

336: Ref? (search throughout methods)

367: Match units.

480: How is whole body P footprint of producing animals pro-rated to that associated with the byproducts?

Reviewer #3 (Remarks to the Author):

What are the major claims of the paper?

The paper presents a comprehensive assessment of phosphorus (P) fluxes associated with fisheries and aquaculture with global scale, and sub-continental resolution. It highlights the importance of Asian economies in reducing the impact of fertiliser application on P retention in aquaculture. The authors identify a turning point in about 1986 after which the flux of P to aquaculture (fertilisation) increased relative to the flux out (i.e. harvest) and attribute much of this change to Asia. From 2004 onwards, globally, these fluxes have caused a net loss of P from land to aquatic ecosystems through aquaculture. The authors identify freshwater finfish aquaculture as a significant driver of P flux from land to aquatic ecosystem.

These two main conclusions have been drawn elsewhere (Bouwman et al., 2013), although the identification of the tipping point appears novel and is used here to inform targets for aquaculture efficiency with respect to sustainable P use, utilising the phosphorus use efficiency (PUE) value, presented in this context by Huang et al. (2019).

I found the paper very interesting to read, the methods were very well presented and documented but at times I questioned the interpretation of the results. In response to the questions posed to reviewers by the Editor, I am left questioning whether (1) the major findings are of sufficient novelty, and (2) whether the model presented represents an improvement on others (e.g. Bouwman et al., 2013). The authors should address these concerns in the paper.

At times the claims appeared contradictory leaving me with doubt as to the interpretations being drawn. For example: Line 300:

“A global P-neutral fishery sector is an idealised target that will be challenging to achieve especially for developing and least developed countries. We could not detect an improving trend in PUE in our data set because of the lack of repeated measurements over time; nevertheless aquaculture is gradually becoming more efficient as modern technology and management play a more important role. China, especially, is playing a leading role by adopting strict regulations (e.g. Regulation on Quality and Safety in Aquaculture, 2003) aiming for a green growth of its aquaculture⁴⁹ although China’s aquaculture is still dominated by environmentally unfriendly practice.”

This statement appears to conflate evidence from data with statements of policy, in a manner that is confusing. I can see little evidence that P use is becoming more efficient in their data (Fig 2) which looks conspicuously like a hockey stick of impending doom. Perhaps the authors could clarify their interpretation here?

Are they novel and will they be of interest to others in the community and the wider field? If the conclusions are not original, it would be helpful if you could provide relevant references.

The authors key objectives (P4 Line 68) were to address the need for estimates of ‘net impact of the global fishery on P flows, improving basic understanding of P biogeochemical cycles and providing support to identify the P management targets.’ The novelty in this objective is in the consideration of all fishery activities, where others have presented similar analyses, apparently using the same data sources, for finfish and seaweed and mollusc data separately (e.g. Bouwman et al. 2011 and 2013), although it is possible that a synthesis is published elsewhere. The authors should use these studies should for comparison with the present study to demonstrate novelty and to validate their approach, where appropriate. For example, the authors indicate that estimating P input values is complicated and that they use the PUE values to infer them. The authors should be clear how their model differs from that described by Bouwman et al (2013), and discuss the implications of these differences with respect to P load estimates. What would be helpful is for the authors to provide a method that improves confidence in the global estimates for P loading (or net P flux from land to aquatic ecosystem) from aquaculture, not simply provides an alternative with little context. Should the methods be sufficiently different, then comparison between estimates will also be of use to the community. The authors should be clear about how this work advances that presented by their colleagues in the field.

The paper provides a useful baseline of P use efficiency in aquaculture and focusses efforts on specific global regions where the biggest gains are to be made.

Please feel free to raise any further questions and concerns about the paper.

Lines 34-44 on land agriculture seems superfluous as does reference to the planetary boundaries, which are fairly contentious in this field at any rate.

Lines 62-75 presents the crux of the paper which should be introduced earlier.

Fig 1: I prefer the more complex fig in sup file to this – it offers greater insight into the model construction. It would be very useful if the authors could also include reference to the data used within this conceptual model.

Line 138: I don't think 'leaching' is the correct word here, perhaps, 'loading'?

Line 146: should be 'at the global scale'.

Line 183-199: other authors have considered more complex scenarios, e.g. the Millennium Ecosystem Assessment scenarios (Bouwman et al., 2013). Why did the authors pick such simple scenarios? Are they realistic?

Line 219-240: this may be overly speculative. Could these measures and their relative effects be summarised in a table instead?

Line 245-279: There are a few sentences in here that need attention. For example: "Historically, human demand for wild fish may indirectly alleviate P shortage on land, especially in countries where wild capture fishery is economically important such as Peru and Japan, or in low income food deficient countries where natural land P resources are strongly limited and P fertiliser too expensive." Having read this a few times I am struggling, still, to find the meaning.

Line 273: actually, it took decades (i.e. 1980s to 2000s) of persistent excessive P loading to switch the global fishery from being an importer of P to land to being a net exporter of P from land – the authors should not underplay the scale (spatial or temporal) of the cause and effect, here.

Line 516: can the use of the PUE approach be validated in some way against another approach, e.g. Bouwman et al (2013).

We would also be grateful if you could comment on the appropriateness and validity of any statistical analysis, as well the ability of a researcher to reproduce the work, given the level of detail provided.

The paper and supplementary files document comprehensively the approach taken. It is a long and complex one that requires the user to accept significant uncertainty in the global data sets being used – as is acceptable in this field.

References:

Bouwman, A.F., A. H. W. Beusen , C. C. Overbeek , D. P. Bureau , M. Pawłowski & P. M. Glibert (2013) Hindcasts and Future Projections of Global Inland and Coastal Nitrogen and Phosphorus Loads Due to Finfish Aquaculture, *Reviews in Fisheries Science*, 21:2, 112-156

Bouwman, A. F., M. Pawłowski, C. Liu, A. H. W. Beusen, S. E. Shumway, P. M. Glibert, and C. C. Overbeek. Global hindcasts and future projections of coastal nitrogen and phosphorus loads due to shellfish and seaweed aquaculture. *Rev. Fisheries Sci.*, 19: 331–357 (2011)

Huang C.L., B. Gao, S Xu, Y. Huang, X. Yan, S. Cui. Changing Phosphorus metabolism of a global aquaculture city. *Journal of Cleaner Production*. 225, 1118-1133

Thanks a lot for helping us to improve our manuscript. We would like to further emphasize novelties of our study, which include at least the following points: (1), we have presented a new data based global quantification of the under-represented yet critical land-aquatic P fluxes driven by human demand for fish; (2) we have found a new and largest pathway that transfers P from aquatic ecosystems to land; (3) we have revealed a shift of the net land-aquatic P transfers, its timing, direction and magnitude and (4) we have provided a target aquaculture phosphorus use efficiency for a future P-neutral fish production section. Land-aquatic P transfers are critical for both food security and environmental sustainability. No studies, to our knowledge, has provided such a full historical and novel perspective on fish-driven P transfers. We revised our manuscript to further emphasize the novelties. We have calculated P budget per unit protein supply for different food production sectors (fish vs. crop + livestock) and have revised the manuscript according to Reviewers' suggestions. We also modified our writing to make sure we did not overstate or oversimplify the conclusions and implications of our research. Please find below our detailed responses to each comment and the rationale behind our revisions. Line numbers are based on the manuscript with tracked changes.

Reviewers' comments:

Reviewer #1 (Remarks to the Author):

This paper nicely shows how the transfer of phosphorus (P) via fisheries has shifted on a global scale over the past several decades. The findings are quite interesting and certainly relevant to global environmental issues, food security and sustainability. Overall the methods seem robust and the paper is well written. I think this will ultimately be an impactful publication, although I do have several suggestions that I think will improve the paper.

Response: Thanks a lot for your positive comments and constructive suggestions. We incorporated your suggestions in the revised manuscript. Details are provided in our responses to your comments below.

One general issue, which I think is common for 'budget' studies such as these, is that it is important that the authors clearly define the spatial scale, i.e., the boundaries and fluxes. For example, the study includes fluxes to and from both marine and freshwater systems, but in several places the authors seem to be most concerned with the oceans, i.e., how much P is delivered to vs. removed from the oceans. I understand that the oceans can be considered the 'end point' of P flows, but from an environmental perspective, net retention of P in freshwaters is just as important as that in marine systems. Furthermore, the emphasis on marine systems could confuse readers, who might think that freshwaters are included in the 'land-human' system (as is sometimes the case for global carbon budgets, for example). Therefore, I ask that the authors 1) clarify that 'aquatic ecosystem' includes both marine and freshwater (e.g., early in the text and in the legend for Fig. 1), and 2) omit statements that could cause confusion. The text on lines 260-263 is a good example of this – why single out ponds? They are aquatic ecosystems too, so in principle human-derived P that is stored in ponds is the same as human-derived P stored in the oceans (in terms of importance, and certainly in terms of the budget).

Response: We added "Here aquatic systems include both marine and freshwater systems." in the main text (line 97) as well as in the legend for Fig. 1. In the result section, we specified "Freshwater aquaculture contributes to most (84-94%) of the P-input, though the share of marine aquaculture increases through time (Figure S8)." (lines 162-164). And we have Figure S8 dedicated to

quantifying the relative contributions from freshwater vs. marine aquaculture. On lines 260-263 (old ms), we used pond as an example. We added “84-94% of the aquaculture P-input went into the freshwater and only a portion could ultimately enter the ocean.” (lines 310-311) to emphasize P-input into freshwater ecosystems.

The authors mention that reliance on ‘fish’ (including shellfish) is likely to increase because of increasing demand for protein. I wonder if they could estimate the relative consequences of reliance on aquatic protein vs terrestrial protein. For example, if the increase in fish harvest since 1950 was replaced by an increase in terrestrial protein, what would happen to the P budget? I realize this is a bit beyond the scope of the paper but nevertheless some quick calculations might be illuminating and would be of great interest.

Response: We calculated total global protein supply from terrestrial sources (from vegetation and livestock) as well as from “fish”. We also searched through literature about the global P-load into aquatic ecosystems from the crop-livestock system. Section 7 of the revised Supplementary Information presents “Comparing aquatic vs. terrestrial P budget (per unit protein)”. When we took capture fisheries and aquaculture together, P loads per unit food protein supply was smaller than that from terrestrial sources. If historical fish P protein had been supplied by the crop-livestock system instead, there would have been more P loaded into aquatic ecosystems. However, when we investigated aquaculture separately, P loading is larger than the crop-livestock system. These results are presented around lines 287-294 in the main text.

Line 95: insert ‘to aquatic ecosystems’ after ‘add’ to make it clear what the fluxes are.

Response: We added these words.

Line 97: is ‘production’ the same as harvest but just different units? (mass for production, P for harvest)?

Response: When we use “production”, we refer to biomass. For “harvest”, sometimes we specified “harvesting wild and cultivated fish” or “P-harvest”. Yes. They refer to the same activity, but sometimes for different subjects and therefore with different units.

Line 160: Seems like Russia/Soviet Union should also be included here.

Response: We added Russia and also Chile and Denmark as examples.

Figure 1 is really nice, and I wouldn’t be surprised to see this reproduced in textbooks in the future.

Response: Thank you.

Figure 4: The units here seem unnecessarily confusing. Just to be clear, a value (in the legend) of 0.01 would mean 1 Tg P yr⁻¹? Why not just relabel scale to Tg?

Response: We modified the figure legend to use Tg P yr⁻¹ as the unit.

Lines 244-247. This is a very confusing sentence. As mentioned earlier, with budget studies like this, it’s really important to keep things clear, and I think this will confuse many readers.

Response: We removed this part.

Lines 286-287. It is curious that efficiency is higher for fish than invertebrates. Fish have much higher body P concentrations than invertebrates. Thus, invertebrates are likely to consume food that has a similar P content to their own bodies, which would seem to render them more efficient than fish at converting food P into body P. Is there a logical explanation for the lower efficiency of invertebrates?

Response: One paper from Vanni et al. (2002) said “With regard to even broader phylogenetic patterns, vertebrates will generally sequester relatively more P in their bodies than invertebrates because of the higher P content of vertebrate bodies”. In their paper, they showed a negative correlation between body P content and P excretion rate (per mass per time, Figure 1 from their paper), that is a higher P use efficiency for species with higher P content. Despite their study focused more on vertebrate aquatic species, their study may serve as an explanation. In this study, we use culture system level PUE, which also depends on farmers’ practice in raising different species and may deviate from a pure physiological perspective. Specific to this sentence, we meant to compare with other major animal protein sources, so we added (e.g., compared to beef and pork) to this sentence.

Vanni, M. J., Flecker, A. S., Hood, J. M. & Headworth, J. L. Stoichiometry of nutrient recycling by vertebrates in a tropical stream: linking species identity and ecosystem processes. *Ecology Letters* 5, 285-293, doi:10.1046/j.1461-0248.2002.00314.x (2002).

Line 292 and nearby: I don’t fully understand where this number (0.84 Tg P yr⁻¹) comes from.

Response: 0.84 Tg P yr⁻¹ is the difference between total harvested fish P and P that was reused as fishmeal and fish oil in year 2010. We rewrote this supplementary section to make it clearer. The new section reads:

“We calculate the amount of harvested P that can be further recaptured by land-based human activities through subtracting from total harvested P by the amount of P that ends up as fishmeal and fish oil. Here, we document the calculation for year 2010, because for this year we found data on global fishmeal and fish oil production as well as the portion used as aquaculture feed. In 2010, total P-harvest is 0.98 Tg P. With 18.5 Tg (live weight equivalent) of captured forage fish being used to produce fishmeal and fish oil in 2010³, and with a global average P concentration of 0.76% from our data, we estimated that 0.14 Tg P ends up in fishmeal and fish oil. Harvested P that could be further recaptured by land activities, excluding fishmeal and fish oil, is therefore 0.84 Tg P (0.98 minus 0.14). We separated P in fishmeal and fish oil into two fractions: one that goes as feed to aquaculture and the other that serves as food/supplementary for human and land animals. In 2010, aquaculture consumed 73% of fishmeal with the rest being fed to poultry and swine and other sectors³. For fish oil, we only found statistics for 2012 when 74% of fish oil was reported to be used by aquaculture with the rest going to direct human consumption or being used in other sectors⁴. We therefore assumed that 73% of fish oil was applied to aquaculture in 2010 to simplify the calculation. The total harvested P that directly goes back into aquaculture through fishmeal and fish oil is 0.10 Tg P (73% x 0.14), leaving 0.4 Tg P being used by human and land animals.”

Line 343: Did calculations use a constant PUE among countries (within a particular culture-system), or was PUE country-(region)-specific?

Response: We did not differentiate PUE by countries within a particular culture-system. We do not have enough PUE data to separate country-level difference in PUE. Aquaculture production is dominated by countries in Asia, where PUE varies widely within each country which does not allow us to detect a statistical difference among countries. We added this to the Methods section (line 626).

Paragraph beginning on line 444. I don’t really understand how uncertainty in body P was explored. What is meant by ‘1 record as the default’? I think this has to do with sample size for body P content for specific taxa, but it’s very hard to follow, even after reading the Supplemental Information.

Response: Yes. It is related to the sample size for body P content. The uncertainty quantification is done using the Monte Carlo method. We randomly sample the P content for each species and then calculate P-harvest budget as presented earlier (lines 545-550). The sample size is different from different species. The range of the body P content covered in our database may not reflect the whole range of body P concentration for under-sampled species, as the literature may be biased towards more economically or scientifically important species. So we borrowed information from higher taxonomic levels to cover a wider uncertainty range especially when the sample size of the corresponding species is small. The criterion to decide when we should borrow information from higher taxonomic levels is when we put ‘1 record as the default’. To make this part clearer, we modified this part and now it reads as:

“We randomly sampled the whole-body P concentration 1000 times for each species in our database and calculated P-harvest budget. The number of compiled whole-body P concentration entries differs for different species. P concentration for each species may not cover the whole range of uncertainty. To avoid under-estimation of uncertainty associated with P concentration, we made use of information from higher taxonomic levels. If the number of P concentration entries were no more than a cut-off criterion (1 record, for example), then the random samples for the corresponding species were generated from the selection pool with P concentration information covered by the taxonomic order which this species belongs to. P budgets were not sensitive to the cut-off criterion that determined the population from which random samples were drawn (see SI: 5), we therefore reported results from setting the cut-off criterion to be 1 entry. When the species level average is different to the average of the taxonomic order level, random samples taking directly from the bigger taxonomic order pool would deviate the average P concentration from the species level pool. In order to keep the same average while covering a reasonable uncertainty range, we conducted normal sampling using the species level average and the standard deviation that is rescaled by the coefficient of variation at the taxonomic order level..... ”

Data: I have a few questions regarding data on body P. First, the authors refer to these data several times, but I cannot find the data. It is not in any tables, and I don't see any data files accompanying the manuscript submission. I can't even find any reference to the data base. It would be nice to have access to these data. Also, what do sample sizes refer to in Fig. S3? The number of studies? The number of individuals? Either way, the sample sizes seem too low, especially for the two invertebrate groups. But this depends on the scope of the analysis. For example, papers by Ikeda contain several hundred measures of body P contents for crustaceans, but most of these are for zooplankton such as copepods. Were these values excluded because zooplankton are not harvested? The authors refer to a 'global' database, but it's not clear if this means global only in a geographic sense, or also in terms of the taxonomic pool being considered. In the end, I doubt that this will affect the conclusions because uncertainty in body P content seems to have little effect on P harvest rates at the global scale. Nevertheless, it would be best to clarify these points, and to include the data.

Response: We uploaded these datasets as supplementary data. We have three .xlsx files named as Data1_Fish_P_Concentration, Data2_Fish_P_Use_Efficiency and Data3_Fish_P_Retention_v1.0. Data1 also covers meta-data related to reference, sources, fish common name, scientific name, family name, order name, habit, experimental type, location, treatment and whether it is wild or raised in addition to whole body P concentration. Data2 covers meta-data related to reference, culture system, source, location, fish common name, scientific name, marine or fresh, fish type and culture system level P use efficiency. And Data3 documents reference, source, raising system type,

location, fish common name, scientific name, experimental treatment and P retention. We double checked this time we have successfully uploaded these data files.

Sample size in Fig. S3 refers to individual experiments. We think it is a decent number of samples (> 1000) that covers 224 species (almost all economically important aquaculture species) across 175 peer-reviewed publications. This database contains whole-body P concentration for each species. A lot of studies in literature only reported P content in fish meat. We only incorporated studies that recorded whole body P concentration. So we needed to filter out a lot of studies. To our knowledge, our study is the first study that collect such a large number of whole body P concentration of fish. Yes. We do not include zooplankton despite there are large samples. Thank you for the constructive suggestions. We added to Fig. S3 “Numbers correspond to individual studies. The entire database covers 224 fish species.” to clarify it.

Supplement, Section 3 (Harvested P to be used by land activities). I understand what this means, but just to be clear, the basic budget (and P-net in particular) did not take this into account, correct?

Response: P-harvest tracks the total P harvested from fish (lines 545-561). A small part of this harvested P ends up as fishmeal and fish oil (SI, section 3). The fraction of fishmeal and fish oil that is applied back to feed aquaculture (here I use P_f in short) is accounted in P-input (lines 603-651). P-net is the difference between P-harvest and P-input. So P_f is an internal exchange for P-net and does not directly affect P-net as long as there is no shift in PUE. However, in practice, an increase in P_f may indirectly affect culture system level PUE through changing the allocation of feeds with different PUEs, therefore, alter P input from other sources, and ultimately affect total P-input and P-net. Other than P_f , the fate of other harvested P is not explicitly tracked. For example, the harvested P may go through human bodies and enter sewage systems, and then end up in water bodies where fish growth may benefit. The fate of these harvested P depends on, for example, the recovery of P in sewage water processing facilities, which differs a lot between cities and countries. We do not track explicitly these secondary transfers as they are indirect fluxes and we do not have enough data to get good quantifications.

Fig S7b: This is very confusing with the different scales and colors. Also the legend doesn't make sense to me.

Response: We modified the legend and separated this panel into 3 panels, with one panel for one fish group. We also modified Fig S8b in a similar way. We think in the new version, Figs. S7 and S8 are easy to understand.

Supplement, Section 7 (Recycling and reuse of harvested P). This section seems like ‘Discussion’ and therefore seems out of place in the Supplement.

Response: We merged this section with the discussion in the “Outlook for a global P-neutral fishery management”. We also moved part of the text that is related to PUE from the result section into the discussion in “Outlook for a global P-neutral fishery management” to make the writing more coherent.

#####

Reviewer #2 (Remarks to the Author):

This paper provides a global review of phosphorous cycling associated with fish and seafood. From the 1950s-1980s, the seafood system was dominated by wild capture, and shifted P from aquatic to terrestrial

systems. More recently, feed-driven aquaculture mean the seafood system brings terrestrial P, in the form of fertilizers, to the aquatic systems. Seafood thus represents a net deposition of .96 Tg P yr⁻¹ into aquatic systems, a meaningful portion of net deposition from all sources.

I am reviewing these results from the perspective of someone knowledgeable about global fish and aquaculture policy discussions, where they aim to have an impact; I'm not a biogeochemist or life cycle analyst. With this lens, I appreciate the scale of the calculation exercise here, but the chosen scope and scale of the analysis does not seem to provide the most important insights. Rather, I am left concerned that it will mislead consumers or other users of this information towards other protein sources, which are in fact worse than the environment.

Response: We appreciate your feedbacks. We did not intend to lead consumers towards other protein sources. Instead, we intended to quantify the global land-aquatic P transfers focusing more on the biogeochemical cycle. We would like to emphasize that human demand for fish plays a unique role in land-aquatic P transfer as it can extract P from the aquatic ecosystems. The extraction perspective is a critical angle that can provide potential solutions to P issues faced by our society, but largely overlooked in literature. Land food production is limited by P availability and thus land food production benefited from P extracted from aquatic ecosystems historically. We intended to raise people's attention towards this unique biogeochemical role of fish production, improve aquaculture P use efficiency and ideally inform management options of fish production which benefit both food supply and environmental sustainability. This is one novel aspect of our study as we view human demand for fish not only as a P polluter, but also a provider of P for land. We compiled a large dataset to reconstruct the return flux of P from aquatic to land ecosystems and demonstrated that this flux is non-trivial. We also compiled phosphorus use efficiency through literature and showed that there is room for improving PUE in the future. The impact of human consumption of fish meat on land-aquatic P transfers depends critically on how we manage our aquatic ecosystems. We intended to advocate for improving PUE instead of misleading consumers towards other protein sources.

To clarify it, and to deal with this insightful concern of the referee, we calculated P load from fish vs. terrestrial protein sources (crop + livestock) per unit of food protein supply, as also suggested by reviewer 1. Section 7 of the revised Supplementary Information presents "Comparing aquatic vs. terrestrial P budget (per unit protein)". When we took capture fisheries and aquaculture together, P loads per unit food protein supply is smaller than that from terrestrial sources. If historical fish P protein would have been supplied by the crop-livestock system, there would be more P loaded into aquatic ecosystems. However, aquaculture alone has a P loading which is larger than the crop-livestock system (lines 287-293). We carefully checked our manuscript to reduce expressions that might mislead consumers. Please check our detailed responses to your comments below.

Major Comments

There are three places where I struggled to understand the importance of this work. First, it is not clear from this article why transfers of P to aquatic systems are a problem. What are the costs of this, and who experiences them? The planetary boundary is referenced, but is the issue really planetary, or is it local? Lines 92-94 say the final disposition of P here isn't the focus of the study, but then why does the study matter? Where is it OK for P to end up?

Response: We put the paper in the context of the P dilemma: on the one hand we are in shortage of P for crop production, and on the other hand we are facing widespread eutrophication which are

primarily driven by excessive P input into water bodies. Anthropogenic activities have resulted in an imbalanced P transfers between land and aquatic systems, characterized by almost unidirectional P transfers, from land to aquatic ecosystems. The main focus of this study is not on why excessive P transfers into aquatic ecosystems are a problem (please refer to, for example, Correll 1998 and Daniel et al., 1998 for this topic), but to quantify the P transfer fluxes from fish production, which was not done before with the level of details we achieved, and a full historical perspective linked to increasing aquaculture. The costs and who experiences them are also not the main focus of this manuscript, but they are valuable questions worth future explorations. We add this point to the discussion (lines 337-338).

We acknowledge that the topic of “planetary boundary” is controversial. A global threshold does not necessarily mean it applies to every place on the Earth. We interpret it as a widespread unsafe condition if the value goes beyond. Take the more accepted concept of global warming as an example. A broad global estimation of warming could not predict whether the temperature will increase or not at a specific location and time. But this broad global estimation does yield important scientific and practical values. From O'Neill et al., (2018), only 44 countries are within this global P boundary if expressed in a per capital level. We think excessive P input into aquatic ecosystems is not only a local issue, but a global one. We cited the per capital number in the revised manuscript to make the number more relevant (lines 35-36).

We do not explicitly track the fate of harvested P as we do not have solid data to correctly separate these fluxes, e.g. the amount of P captured in sewage depends on facility types and countries. We did estimate the harvested fish P that ends up as fishmeal and fish oil for one year in which we could find all data needed and we extended the discussion section (SI, section 7; Main text, lines 388-392). As a first step, it is valuable to provide an overall quantification of the fluxes. We hope this study will attract people's attention to help collect more relevant data and to guide best management of P resources.

It is okay for P to end up, for example, in the crop that serves as human food. However, the question ‘Where is it OK for P to end up?’ is not trivial to answer as elements are constantly cycling, and thus will not permanently end up in one location. The answer will depend on the time scale we look into it and thus the answer will differ for different communities.

O'Neill, D. W., Fanning, A. L., Lamb, W. F. & Steinberger, J. K. A good life for all within planetary boundaries. *Nature Sustainability* 1, 88-95, doi:10.1038/s41893-018-0021-4 (2018).

Correll, D. L. The role of phosphorus in the eutrophication of receiving waters: A review. *Journal of Environmental Quality* 27, 261-266, doi:10.2134/jeq1998.00472425002700020004x (1998).

Daniel, T. C., Sharpley, A. N. & Lemunyon, J. L. Agricultural phosphorus and eutrophication: A symposium overview. *Journal of Environmental Quality* 27, 251-257, doi:10.2134/jeq1998.00472425002700020002x (1998).

Second, it is odd to me to scope the study to combine capture fisheries and aquaculture, as if there is a natural balance in the two processes simply because they both result in fish in the market. Why does capture fisheries' net terrestrial shift only offset aquaculture's net aquatic shift, rather than the net aquatic shift driven by intensive food production for all proteins? What if P contributions could be more effectively reduced through other proteins? Rather, the importance to highlight seems to be shifting from

capturing wild fish which use natural ambient P, to application of chemical fertilizers, which are much different processes. Each of these sources might be of interest as components of the food system level, but it is not clear that I've learned something about the state of the world by aggregating the two, to the exclusion of other methods of food production.

Response: We put fisheries and aquaculture together as they are both driven by human's demand for fish from aquatic ecosystems. We think they are well defined in terms of market, economy, social recognition, product and food grow environment, etc. For example, the Food and Agriculture Organization of the United Nations (FAO) reported capture fisheries and aquaculture fish production together through the Global Fishery Production database, not together with crop and livestock. In addition to Fig. 2 where we reported the overall budget of P-harvest, P-input and P-net, we also provided Fig. S7 where we showed P-harvest, separating wild capture and aquaculture. In this way, we provide information on both aggregating and separating wild capture and aquaculture.

Historically, capture fisheries' net terrestrial shift did not only offset aquaculture's net P input into aquatic, but also the net aquatic P input caused by terrestrial food production, as it is shown by the positive P-net in our Fig.2. At the current stage, P-harvest is not enough to offset P-input from aquaculture. As a first step, we set a future goal to reach a zero P-net flux. A more ambitious goal is to go beyond the zero P-net and to also offset P load from the crop-livestock food production system. We think it is more logical to discuss whether we can double current PUE to reach 48% by 2050 in order to have a zero P-net flux from capture fisheries and aquaculture first. If we can reach this goal easily, then the next step is to go further to investigate how much our demand for fish meat can offset the P load from the crop-livestock food production system into aquatic environment. We hope this manuscript can raise attention on aquaculture management to improve PUE and to push aquaculture towards the direction of offsetting terrestrial's P input into aquatic.

Third, such environmental impact studies are frequently taken in the fish literature to scare consumers away from fish, depicting it as environmentally unfriendly. However, this is only good for the environment if what consumers would eat instead has a lower P footprint. Thus, some additional context is necessary to compare numbers reported here with what is known about P from other protein sources, or at least to be circumspect about the fact that reducing P from aquaculture might lead to more P someplace else in the food system.

Response: We now added a section in the supplementary information on calculating P load into aquatic ecosystems per unit food protein supply from different food production sectors. We discussed in the main text that "If historical fish P protein had been supplied by the crop-livestock system, there would have been more P loaded into aquatic ecosystems". Please also refer to our response to your first general comment.

If there's all this runoff feeding aquatic and marine systems, isn't it helpful to extract it (by catching fish)?

Response: One unique point of our study is that we showed that fish catching transferred a significant P flux that goes from aquatic to terrestrial ecosystems. We showed that this flux is bigger than any other known fluxes that transfer P from aquatic to terrestrial ecosystems. This is a novel point we intended to emphasize in our study. It is desirable to make good use of runoff that is rich in nutrients to raise fish and then extract it by catching fish. It shares a similar idea as the integrated aquaculture and agriculture systems (IAA, Costa-Pierce, 2002), where wastes from agriculture serve as food for fish, and harvested fish or fish wastes act to fertilize crops and feed livestock. In practice, it requires good managements and large efforts to effectively recycle nutrients as mentioned in the main text when we talk about IAA (Lines 360-380). Nevertheless, we

agree with the Reviewer that we should pay attention to the role of capture fisheries and aquaculture in returning P back into the terrestrial ecosystems and this is what we stated in lines 381-382.

Costa-Pierce, B. Ecological aquaculture: the evolution of the blue revolution. (2002).

Minor Comments

Does fish mean only finfish, or also invertebrates?

Response: In this study, “Finfish, crustaceans and mollusks, hereafter generalized as “fish” (lines 51-53). This type of generalization is also a convention in FAO reports.

32: having trouble reconciling at “safe global” level with preceding discussion of observed local saturation. What does a global number mean?

Response: We acknowledge that the topic of “planetary boundary” or “safe global” is controversial. A global threshold does not necessarily reflect local conditions. In the case for “planetary boundary”, we interpret it as a widespread unsafe condition if the value goes beyond. We acknowledge that local studies would provide meaningful information under specific conditions. We also think this kind of global studies are valuable in a certain way as it provides broadscale information. We keep “planetary boundary” for the context of this study as it exists in current science literature. Please also refer to our response to your first major comment.

54: This is a very broad statement to make about aquaculture in particular, separate from agriculture.

Response: We added “along with growing sustainability concerns on other food production sectors” (lines 61-62) to say other food production sectors also raise concerns about sustainability.

61: What is considered state-of-the-art?

Response: As aquaculture is concentrated in Asia and China takes the leading share in aquaculture fish production, we think it is pertinent to use China, not the developed countries, as an example to show the value of PUE. The paper we cited was published in 2015 and is relative recent. It is a synthetic research on previous studies focusing on China. If the referee’s question is whether values we cited are representative, we think the answer is yes.

89: This is feed and fertilizers specific to producing fish, yes?

Response: In this study, “Finfish, crustaceans and mollusks, hereafter generalized as “fish”” (lines 51-53). So P from feeds and fertilizers include all P that are used to purposely raise these species through feeds and fertilizers.

Fig 1: Fish aren’t fertilized P directly, right? This goes to plant-based feed components. Also, harvested fish should link to fish feed, through meal. This diagram should be much clearer about the transport pathways.

Response: We moved a complex version of Fig. 1. from the supplementary information to the main text as the new Fig. 1. The purpose of fertilizer is to boost aquatic primary productivity to increase food supply (e.g., algae, plant) especially for herbivores and omnivores. In the main text, we explained “through fertilizers that enhance the primary productivity of aquatic ecosystems (e.g., for herbivorous and omnivorous species).” (lines 64-65).

119: “P-harvest” is P from capture fisheries only?

Response: P-harvest include P harvested from both capture fisheries (wild) and aquaculture. We added “(wild + aquaculture)” to make it clearer. In the introduction and method sections, we have explained what P-harvest refer to.

123: Again, this is just farming for aquaculture feed? Needs to be clearer what this is.

Response: we added (aquaculture feed and fertilizer) when we mention P-input. In the introduction and method sections, we have explained what P-input refer to.

124-128: This is hard to unpack.

Response: We rewrote here as “P-harvest estimated here is the largest known pathway that transfer P from aquatic ecosystems to land, compared to 0.0056 Tg P yr⁻¹ from anadromous (migratory) fish¹, 0.099 Tg P yr⁻¹ from seabird colonies² and 0.16 Tg P yr⁻¹ from sea salt deposition³. 99% of P-harvest came from wild capture (mostly marine capture) in 1950, and this share decreased to 62% in 2016 (Figures S7, S8).”

137: This might be one of comparisons I asked about above, but I cannot tell exactly what is being said here. This is runoff losses?

Response: we rewrote this part as “In comparison, leaching, runoff and erosion losses of fertilizer P from croplands³⁷ to freshwater were reported to be 0.6 Tg P yr⁻¹ from Mekonnen and Hoekstra (2018) over 2002-2010 and Lun et al. (2018) reported a larger P loss rate of 3.7 Tg P yr⁻¹ from croplands to water bodies through runoff over the same period.”

141-43: I cannot tell what this says.

Response: The first sentence here describes the percentage contributions to P-input from different fish groups (finfish, crustaceans and molluscs) when we take freshwater aquaculture as the boundary to calculate the total. And the second sentence describes the percentage contributions to P-input from different fish groups (finfish, crustaceans and molluscs) when we take marine aquaculture as the boundary to calculate the total. We report results separately for freshwater vs. marine aquaculture to give more details as also suggested by another referee. We realize it was not clear here as Figure S8 was complex. We redrew Figure S8. We modified the legend and separated Fig S8b into 3 panels, with one panel for one fish group. We think in the new version, the figure and text together make this part easier to understand.

189: This feels like a “scare” number that may make little sense practically. “Business as usual” sounds like the bad IPCC scenario, but that might not apply here. In particular, if people don’t eat aquacultured fish, what will they eat, and will that be better for P? Also, what are the incentives for feed producers to increase PUE? Is there a reason to think it would be unchanged? If this can’t be done sensibly, it should be removed from the paper.

Response: We use “Business as usual” to show if we do not improve PUE, how much P-net we will get with the future growing need for aquaculture fish. We use this scenario as we think it is easy for people to understand as a lot of readers might be familiar with the IPCC scenario. The purpose here is not to say we will do nothing in reality. We also have a scenario in which people will take action, and we calculated the goal in improving PUE if we want, as least a P-neutral fish production. We think it is useful to estimate these numbers under “if” conditions. We add the discussion about

P load from other food production sectors and compared them on a per unit protein supply scale. Please also check our response to your previous comments.

196: This assumes variation in PUE driven by heterogeneity in the efficiency of using a single technology, rather than heterogeneity in technologies used to produce various species in different environments. Does this really provide a good indication of potential PUE?

Response: As PUE and PRE data are compiled from different aquaculture production systems and we use a range instead of a single value, these upper values represent multiple technologies, instead of one single technology. Efficient aquaculture systems are certainly not diverse enough to cover all aquaculture practice, but they demonstrated the capacity for improvements under current technology and practice.

244: Not clear exactly what is being counted here, especially with such a huge range, and probably considerable reference year dependence.

Response: We redid a literature search and wrote this part as “The total P load into aquatic ecosystems from crop-livestock is reported to be 4 Tg P/yr (fertilizer and manure sources only) in 2000 from Bouwman, et al. (2013), 5 Tg P /yr (fertilizer and manure), or 13.5-25 Tg P/yr if land use change was additionally accounted from Peñuelas et al. (2013) over 2005-2011, 12.9 Tg P from Chen and Graedel (2016) in 2013, and 9.7 TgP /yr from Lun et al. (2018) during 2002-2010.”

246: How does this lead to double counting? Does this arise from where things are measured, so they might be utilized “downstream” of the initial point of measurement?

Response: We meant to say P lost from crop-livestock systems may function as aquaculture feed. We realized that the logic here did not need this sentence as we only compared the budget. So we deleted this expression.

252: Is this gross or net of utilization?

Response: We do not understand what the comment refer to. Here we refer to only the input side.

275: Again unclear on what the planetary boundary means in this application. What will this look like?

Response: Please refer to our response to your first major comment.

281: This is an odd goal. Why credit aquaculture with the benefits from capture fisheries? Shouldn't the goal be to reduce pollution from each form of production to a level where the benefits of marginal abatement equal the marginal costs?

Response: Please refer to our response to your second major comment.

283: It is not obvious to me why net landward is good. Alternatively, how much would global capture have to increase?

Response: We put our study under the context of P limitation for terrestrial food production and widespread eutrophication with excessive P input into aquatic ecosystems. We think landward P transfer is good for land and removal of excessive P out of aquatic ecosystems is good for aquatic ecosystems. The global capture of fish is driven by human demand for fish meat. The historical global fish production is based on FAO and the Sea Around Us (<http://www.seaaroundus.org/>) databases, while the future increase of aquaculture fish production and stabilization of wild capture are from the projection that considers assumptions of fish supply and demand, health of wild

fisheries, fish prices, human population growth, GDP growth, and technological progress (Waite et al., 2014).

Waite, R. et al. Improving productivity and environmental performance of aquaculture. (2014).

289: But total P-input might not, if people are eating fish instead of more P-intensive proteins.

Response: Thank you for this insightful point. We removed this sentence and wrote this discussion section by merging part of the supplementary information and part of the content in the result section. Specifically, we compared between P load from fish vs. crop + livestock sectors on a per unit protein supply base to reduce the confusion here.

301: Is this true? Many least developed countries rely more on capture than aquaculture.

Response: We removed this sentence.

310: Not necessary?

Response: We removed this sentence.

336: Ref? (search throughout methods)

Response: We do not understand what did the reviewer mean by this comment.

367: Match units.

Response: We changed million tonnes to Tg.

480: How is whole body P footprint of producing animals pro-rated to that associated with the byproducts?

Response: We collected data on whole body P content of different fish species. Depending on the specific studies that measured fish P content, some studies measured P concentrations from different fish organs and calculated the integral P content taking into account weights of different organs, and other studies extracted P from the whole fish. For animals other than fish, we did not collect data for whole body P content data. Tacon, et al., 2009 provided a broad estimation of P content from different feeding sources.

Tacon, A. G. J., Metian, M. & Hasan, M. R. Feed ingredients and fertilizers for farmed aquatic animals: sources and composition. FAO Fisheries and Aquaculture Technical Paper. No. 540. Rome, FAO. 2009. 209p. (2009).

#####

Reviewer #3 (Remarks to the Author):

What are the major claims of the paper?

The paper presents a comprehensive assessment of phosphorus (P) fluxes associated with fisheries and aquaculture with global scale, and sub-continental resolution. It highlights the importance of Asian economies in reducing the impact of fertiliser application on P retention in aquaculture. The authors identify a turning point in about 1986 after which the flux of P to aquaculture (fertilisation) increased relative to the flux out (i.e. harvest) and attribute much of this change to Asia. From 2004 onwards, globally, these fluxes have caused a net loss of P from land to aquatic ecosystems through aquaculture.

The authors identify freshwater finfish aquaculture as a significant driver of P flux from land to aquatic ecosystem.

These two main conclusions have been drawn elsewhere (Bouwman et al., 2013), although the identification of the tipping point appears novel and is used here to inform targets for aquaculture efficiency with respect to sustainable P use, utilising the phosphorus use efficiency (PUE) value, presented in this context by Huang et al. (2019).

I found the paper very interesting to read, the methods were very well presented and documented but at times I questioned the interpretation of the results. In response to the questions posed to reviewers by the Editor, I am left questioning whether (1) the major findings are of sufficient novelty, and (2) whether the model presented represents an improvement on others (e.g. Bouwman et al., 2013). The authors should address these concerns in the paper.

Response: Thank you for your time and positive comments. We would like to further emphasize novelties of our study, which include at least the following points: (1), we presented a data based global quantification of the under-represented yet critical land-aquatic P fluxes driven by human demand for fish; (2), we found a new and largest pathway that transferred P from aquatic ecosystems to land; (3), we revealed a shift of the net land-aquatic P transfers, its timing, direction and magnitude; (4), we provided a target aquaculture phosphorus use efficiency for a future P-neutral fish production section. We strengthened these novel points in the revised manuscript. For example, we added sentences like “Through a new data based global quantification of the under-represented yet critical land-aquatic P fluxes, we found a new and largest pathway that transferred P from aquatic ecosystems to land.....” and “We also revealed, for the first time, a shift of the net land-aquatic P transfers driven by human demand for fish”.

Our study is different from Huang et al. (2019) spatially as they only focused on one location (city), Zhangzhou in China, while our study covers the globe. We do not know why there is “global” in their title. They showed that improving PUE is urgent based on the large P loading into aquatic environment, while our study proposed baseline PUE, future PUE target and potential zoom for improving PUE through analyzing a wide range of aquaculture systems. We added this reference to the main text while we talked about aquaculture PUE in China (Line 375). For the comparison with Bouwman et al., 2013, please refer to our specific responses to your comments below.

At times the claims appeared contradictory leaving me with doubt as to the interpretations being drawn. For example: Line 300:

“A global P-neutral fishery sector is an idealised target that will be challenging to achieve especially for developing and least developed countries. We could not detect an improving trend in PUE in our data set because of the lack of repeated measurements over time; nevertheless aquaculture is gradually becoming more efficient as modern technology and management play a more important role. China, especially, is playing a leading role by adopting strict regulations (e.g. Regulation on Quality and Safety in Aquaculture, 2003) aiming for a green growth of its aquaculture⁴⁹ although China’s aquaculture is still dominated by environmentally unfriendly practice.”

This statement appears to conflate evidence from data with statements of policy, in a manner that is confusing. I can see little evidence that P use is becoming more efficient in their data (Fig 2) which looks conspicuously like a hockey stick of impending doom. Perhaps the authors could clarify their interpretation here?

Response: From our data, we could not detect an improvement in PUE through time. We expect the aquaculture would become more efficient in the future in countries like China taking strict regulations on aquaculture. We rewrote this sections as “We could not detect an improving trend in PUE in our data set because of the lack of repeated measurements over time. Nevertheless, we are optimistic in aquaculture becoming more efficient with modern technology and management playing a more important role in future.....” (lines 350-353), followed by the paragraph discussing the modern technology. We carefully modified the manuscript to reduce the ambiguity like this one.

Are they novel and will they be of interest to others in the community and the wider field? If the conclusions are not original, it would be helpful if you could provide relevant references.

Response: Please see our responses to your previous comments about the novelties of this study. We added the references of Huang et al. (2019) and Bouwman et al. 2011 and 2013

The authors key objectives (P4 Line 68) were to address the need for estimates of ‘net impact of the global fishery on P flows, improving basic understanding of P biogeochemical cycles and providing support to identify the P management targets.’ The novelty in this objective is in the consideration of all fishery activities, where others have presented similar analyses, apparently using the same data sources, for finfish and seaweed and mollusc data separately (e.g. Bouwman et al. 2011 and 2013), although it is possible that a synthesis is published elsewhere. The authors should use these studies should for comparison with the present study to demonstrate novelty and to validate their approach, where appropriate. For example, the authors indicate that estimating P input values is complicated and that they use the PUE values to infer them. The authors should be clear how their model differs from that described by Bouwman et al (2013), and discuss the implications of these differences with respect to P load estimates. What would be helpful is for the authors to provide a method that improves confidence in the global estimates for P loading (or net P flux from land to aquatic ecosystem) from aquaculture, not simply provides an alternative with little context. Should the methods be sufficiently different, then comparison between estimates will also be of use to the community. The authors should be clear about how this work advances that presented by their colleagues in the field.

Response: Our study focuses on the global land-aquatic P budget and is data-driven. As land-aquatic P transfers are very important for both food and environmental sustainability, there are a lot of local studies quantifying P budgets. However, there are not many global scale studies. In addition to studies from Bouwman et al. 2011 and 2013 that quantifying large scale P loads of aquaculture, Hall et al., 2011 is the other study reporting on aquaculture P budget. Firstly, we argue that these studies are less data driven as they rely on model parameters and assumptions related to these parameters. Bouwman et al (2013) aggregated aquaculture finfish species into 15 major groups and assigned a set of parameters for each major group. These parameters are inferred (or calibrated) from data, but may not well represent aquaculture practice due to the highly diverse nature of aquaculture and the limited data references. For example, in Bouwman et al. 2013, the feed conversion ratio (FCR, P in feed per unit P gain in fish) of non-compound feed is 6 for all fish groups except the Tuna, bonito and billfish group (with a value of 10). In practice, FCR is likely to vary among fish species, feeding material, feeding practice etc. We think our data driven approach is more likely to reflect the real aquaculture practice than earlier modelling studies. Secondly, we would like to emphasize on the P-harvest flux and the landward P transfers during the historical period. We showed in this study, that historically, fish harvest transferred a significant amount of P from aquatic to terrestrial ecosystems, and is the biggest P pathway that brings P from aquatic to land systems. We think this novel finding is scientifically important as this flux is non-trivial. Bouwman et al. 2013 focused more on the P-input through aquaculture, and less

attention was paid to the P-extraction perspective. Our finding of historical P-extraction relies on our new whole body P concentration database. No other studies, to our knowledge, have put such an effort to reconstruct historical P budget driven by human demand for fish. For other novelties of our study, please refer to our response to your overall comment.

We added the discussion with regard to the aquaculture P-input in our study vs. that from Bouwman et al. 2011 and 2013 in lines 306-310: “The higher aquaculture P-input from our data driven estimation compared to the modeling result of Bouwman, et al. (2013) and Bouwman, et al. (2011) can partly be explained by the fate of P after it enters the aquatic ecosystems. For example, Bouwman et al. (2013) quantified P release from ponds and assumed that particulate P were not released from pond systems.” In the Method: Overview part we also mentioned the difference between our study and Bouwman, et al. (2013)

Hall, S. J., Delaporte, A., Phillips, M. J., Beveridge, M. & O’Keefe, M. *Blue Frontiers: Managing the Environmental Costs of Aquaculture. The WorldFish Center, Penang, Malaysia (2011).*

The paper provides a useful baseline of P use efficiency in aquaculture and focusses efforts on specific global regions where the biggest gains are to be made.

Please feel free to raise any further questions and concerns about the paper.

Response: Thank you for the positive comments.

Lines 34-44 on land agriculture seems superfluous as does reference to the planetary boundaries, which are fairly contentious in this field at any rate.

Response: we acknowledge this controversy in the field. Please refer to our response to the major point raised by the second reviewer with regard to the planetary boundary. Two reviewers suggested to compare P budget between fish protein production sector and terrestrial protein production (vegetation + livestock). We think it is better to provide the context here.

Lines 62-75 presents the crux of the paper which should be introduced earlier.

Response: We introduced the broad scale context first and then went into the specific issues with regard to fish P budget. We agree with the Reviewer that it will gain more attention if we put the issue even earlier. We think our current order of presentation is also logical and easy for the readers to follow.

Fig 1: I prefer the more complex fig in sup file to this – it offers greater insight into the model construction. It would be very useful if the authors could also include reference to the data used within this conceptual model.

Response: We replaced this figure with the complex version, and referred to the methods section in which we gave details about data sources.

Line 138: I don’t think ‘leaching ‘ is the correct word here, perhaps, ‘ loading’?

Response: We changed.

Line 146: should be ‘ at the global scale’.

Response: We changed into “Continentally”

Line 183-199: other authors have considered more complex scenarios, e.g. the Millennium Ecosystem Assessment scenarios (Bouwman et al., 2013). Why did the authors pick such simple scenarios? Are they realistic?

Response: We argue that the objectives of the Millennium Ecosystem Assessment scenarios (Bouwman et al., 2013) and those of our study are different. We use the scenarios to derive the PUE level we should reach in order to have a P-neutral fish production system, whereas Bouwman et al., 2013 derived P release from aquatic ecosystems if some of the model parameters is increased or decreased by a certain value under assumptions. As mentioned earlier, Bouwman et al., 2013 is a model driven study. A more complex model does not necessarily yield to a better prediction especially when there are not enough data to support. In the case of predicting aquaculture P-input, supporting data are limited and associated with large uncertainties. In this situation, we think a simple scenario is more straightforward for readers to understand. For future fish biomass production, our projection from Waite et al., 2014 also takes into account fish supply and demand, health of wild fisheries, fish prices, population growth, GDP growth, and technological progress, which is not necessarily less realistic compared to Bouwman et al., 2013. To make it clearer, we added “after accounting for fish supply and demand, health of wild fisheries, fish prices, population growth, GDP growth, and technological progress” in the fish production projection.

Waite, R. et al. Improving productivity and environmental performance of aquaculture. (2014)

Line 219-240: this may be overly speculative. Could these measures and their relative effects be summarised in a table instead?

Response: Here we used information from upper PUE and PRE to show that current technology is capable of achieving the aquaculture PUE of 48%. We used the 95 and 75 percentiles instead of a single maximum value. We calculated the average PUE if we use the 75 percentile PRE values and the historical average relative contributions of P gains from finfish, crustacean and molluscs species. The value reached 48%. It is a strong signal to demonstrate that current technology can bring the aquaculture P use efficiency to the 48% if everything else is not considered. We wrote this part as “Finfish dominated the farmed fish production and contributed to 87% of total P in harvested fish (mean, 1950-2016), while the share of crustacean species grew from near zero in 1950 to 6% in 2016 due to the decline in cultivating molluscs (Fig. S7). If we assume a PRE of 21% from raising molluscs and with 87% of harvested fish P from finfish, the 75th percentile of PREs would correspond to an aquaculture PUE of 48%.”. Fig. S7 shows the relative contributions from different fish groups.

Line 245-279: There are a few sentences in here that need attention. For example: “Historically, human demand for wild fish may indirectly alleviate P shortage on land, especially in countries where wild capture fishery is economically important such as Peru and Japan, or in low income food deficient countries where natural land P resources are strongly limited and P fertiliser too expensive.” Having read this a few times I am struggling, still, to find the meaning.

Response: Here we emphasized the importance of P transfer from aquatic to land ecosystems through harvested fish. This flux is frequently overlooked in P quantification. However, from our calculation, this flux is the largest pathway that return P from aquatic to terrestrial ecosystems. We rewrote this part as “Historically, human demand for fish may indirectly alleviate P shortage on land as wastes associated with fish processing or consumption would serve as crop fertilizer or feeds for livestock. Compared to current known pathways that transfer P from aquatic to land systems

(0.0056-0.16¹⁻³ Tg P yr⁻¹), fish driven landward P flux was the biggest yet largely overlooked, at least in the 1980s.”

Line 273: actually, it took decades (i.e. 1980s to 2000s) of persistent excessive P loading to switch the global fishery from being an importer of P to land to being a net exporter of P from land – the authors should not underplay the scale (spatial or temporal) of the cause and effect, here.

Response: We modified our expressions to be more objective on the cause and effect.

Line 516: can the use of the PUE approach be validated in some way against another approach, e.g. Bouwman et al (2013).

Response: Our PUE based P-input is comparable with the model estimation from Hall et al., 2011 through multiplying estimated farming area by the parameter representing per unit area nutrient and feed input (lines 135-138). Our data-based estimation of P-input is not directly comparable with the modeling study of Bouwman et al (2013). We quantified P load into the aquatic ecosystems, while Bouwman et al (2013) estimated P release from aquaculture into other water bodies based on assumptions on P retention in different aquaculture systems. We added the discussion with regard to the aquaculture P-input in our study vs. that from Bouwman et al. 2011 and 2013 in lines 311-316: “The higher aquaculture P-input from our data driven estimation compared to the modeling result of Bouwman, et al. (2013) and Bouwman, et al. (2011) can partly be explained by the fate of P after it enters the aquatic ecosystems. For example, Bouwman et al. (2013) quantified P release from ponds and assumed that particulate P were not released from pond systems, while our study quantified the total P-input into aquatic systems that include ponds” Please also refer to our responses to your previous comments related to Bouwman et al (2013).

We would also be grateful if you could comment on the appropriateness and validity of any statistical analysis, as well the ability of a researcher to reproduce the work, given the level of detail provided.

The paper and supplementary files document comprehensively the approach taken. It is a long and complex one that requires the user to accept significant uncertainty in the global data sets being used – as is acceptable in this field.

Response: Thank you. All new data on fish P content and PUE have been made freely available so that they can support further research analysis

REVIEWERS' COMMENTS:

Reviewer #1 (Remarks to the Author):

The authors have done a nice job addressing my comments and I am generally satisfied with the revision. I have a few minor points:

Page 4: When defining phosphorus use efficiency here, I think it would be clearer to say 'We define PUE as P harvested via fish biomass divided by P input via feed and fertilizer. PUE was calculated at the farm level.' And then go into the details that follow the current sentence. I think clearly stating the equation here is more transparent and allows the reader to understand without having to look at the Methods.

In the Fig 1 legend, state that the dashed grey arrows are 'complex interactions...' (as you state clearly in Fig S1).

Page 7, around lines 140-150 when comparing P harvest and P input with other fluxes, it would be informative to include global P transport from rivers to oceans (not just transport to freshwaters). Bennett et al (2001; Bio Science) give a flux of 22 Tg P/yr from land to oceans via rivers. This flux is a lot higher than any of those mentioned. I am not sure if this value has been revised since the Bennett al paper, which is nearly 20 years old.

Fig 4 units can still be clarified better. If I interpret this correctly it's (Tg P in the 1980s minus Tg P in the recent decade)/(yrs of time elapsed). Is that correct? I would state this clearly in the legend. And if you can specify the years involved (i.e., the denominator) that would be even better.

Michael J Vanni

Reviewer #2 (Remarks to the Author):

Thank you for your thoughtful response to my concerns.

Reviewer #3 (Remarks to the Author):

I thank the authors for addressing the reviewers comments - you have done a good job. I offer below some general and specific corrections. I have identified some passages that were not easy to follow and I hope you can work with the editorial team to address these, prior to your manuscript being accepted for publication.

I would also request the following revisions be made:

line 34: load from/to what?

passage 132-134: some confusing passages

Passage 149-153: some confusing passages

Figure 3: why is there a large spike in Eastern Europe in the 1980s?

Line 239-240: confusing end of sentence

Line 247: what is 'residues P'?

Passage 258-282: some confusing passages

Line 260: define 'region'

Line 261: perhaps here to balance your statement you could refer to large literature on internal loading and make the point that aquaculture operating in natural lakes may result in a large legacy P problem, potentially extending poor water quality for decades following changes in practices (Sharpley et al., 2013: <https://dl.sciencesocieties.org/publications/jeq/abstracts/42/5/1308>).

Line 317: define 'recently'

Line 342-343: some issues here - not completely clear what you mean by 'fish P' or 'P use..'

Line 355: 'reserches'?

REVIEWERS' COMMENTS:

Reviewer #1 (Remarks to the Author):

The authors have done a nice job addressing my comments and I am generally satisfied with the revision. I have a few minor points:

Reply: Thank you for your positive comments.

Page 4: When defining phosphorus use efficiency here, I think it would be clearer to say 'We define PUE as P harvested via fish biomass divided by P input via feed and fertilizer. PUE was calculated at the farm level.' And then go into the details that follow the current sentence. I think clearly stating the equation here is more transparent and allows the reader to understand without having to look at the Methods.

Reply: We modified here “We define PUE as P harvested via fish biomass divided by P input via feed and fertilizer. PUE are calculated at the farm level” as suggested by the reviewer to make it easier for readers.

In the Fig 1 legend, state that the dashed grey arrows are 'complex interactions...' (as you state clearly in Fig S1).

Reply: We added dashed grey arrow to the Fig caption.

Page 7, around lines 140-150 when comparing P harvest and P input with other fluxes, it would be informative to include global P transport from rivers to oceans (not just transport to freshwaters). Bennett et al (2001; Bio Science) give a flux of 22 Tg P/yr from land to oceans via rivers. This flux is a lot higher than any of those mentioned. I am not sure if this value has been revised since the Bennett al paper, which is nearly 20 years old.

Reply: We added “For reference, 4-22 Tg P yr⁻¹ are transported from rivers to oceans (Ref³⁸: 4 Tg P yr⁻¹; Ref³⁹: 9 Tg P yr⁻¹; Ref⁴⁰: 22 Tg P yr⁻¹).” to the main text. We searched through literature that quantified P transfer from river to ocean. The budget varies among studies. We cited 3 studies that represent the range of estimations.

Fig 4 units can still be clarified better. If I interpret this correctly it's (Tg P in the 1980s minus Tg P in the recent decade)/(yrs of time elapsed). Is that correct? I would state this clearly in the legend. And if you can specify the years involved (i.e., the denominator) that would be even better.

Reply: Yes. We added to the Figure $\log(\text{mean}_{2007-2016} - \text{mean}_{1980-1989})$ to help readers better understand the metric.

Reviewer #2 (Remarks to the Author):

Thank you for your thoughtful response to my concerns.

Reply: Thank you for your positive comments.

Reviewer #3 (Remarks to the Author):

I thank the authors for addressing the reviewers comments - you have done a good job. I offer below some general and specific corrections. I have identified some passages that were not easy to follow and I hope you can work with the editorial team to address these, prior to your manuscript being accepted for publication.

Reply: Thank you for your positive comments. We modified the text to make the manuscript clearer.

I would also request the following revisions be made:

line 34: load from/to what?

Reply: We added to the ocean (mostly from the land) to make it clearer.

passage 132-134: some confusing passages

Reply: We followed the editorial suggestions to remove words like “novel”, “new” and rewrite the sentence as “P-harvest estimated here is the largest pathway that transfer P from aquatic ecosystems to land, compared to currently known pathways, i.e., 0.0056 Tg P yr⁻¹ from anadromous (migratory) fish³⁴, 0.099 Tg P yr⁻¹ from seabird colonies³⁵ and 0.16 Tg P yr⁻¹ from sea salt deposition” to make it easier to understand.

Passage 149-153: some confusing passages

Reply: we rewrite here as “Within the freshwater aquaculture, most of P-input ends up in raising finfish (95–100%), and the share from raising crustaceans slightly increases with time, to reach 5.31% in 2010. In the marine aquaculture, most of P-input (> 90%) ends up in finfish aquaculture during 1950–1970; however, after 1990, around 50% of marine aquaculture P-input goes into raising crustacean species (Supplementary Fig. 8).”

Figure 3: why is there a large spike in Eastern Europe in the 1980s?

Reply: The spike is related to how we treat Soviet Union. Previously we did not put Soviet Union in the plot. We modified this plot by incorporating the contribution from Soviet Union and put a note in the caption of the figure saying: “Note Soviet Union is assigned as part of East Europe before its collapse and countries are assigned into different continents according to their geo-location after the collapse.”

Line 239-240: confusing end of sentence

Reply: We rewrite here as “The global fishery P input into aquatic ecosystems reached 2.06 Tg P yr⁻¹ in 2016, which is significant despite smaller than P load through crop-livestock.” to reduce confusing.

Line 247: what is 'residues P'?

Reply: We removed “wastes or residues” here.

Passage 258-282: some confusing passages

Reply: We are not sure about the location of the unclear part the reviewer refers to.

Line 260: define 'region'

Reply: we changed ‘region’ to ‘local’.

Line 261: perhaps here to balance your statement you could refer to large literature on internal loading and make the point that aquaculture operating in natural lakes may result in a large legacy P problem, potentially extending poor water quality for decades following changes in practices (Sharpley et al., 2013: <https://dl.sciencesocieties.org/publications/jeq/abstracts/42/5/1308>).

Reply: we followed the reviewer’s suggestions and added “Such internal P loading, if not managed properly, would result in the legacy P problem that might extend poor water quality issues for decades after adopting good management practice⁵⁰ .”

Line 317: define 'recently'

Reply: We changed “recently” to “during the recent decade”

Line 342-343: some issues here - not completely clear what you mean by 'fish P' or 'P use..'

Reply: We rewrite this part as “These technologies are promising to recover most P transferred from the aquatic to land-human systems through harvested fish in future” **to make it less confusing.**

Line 355: 'reserches'?

Reply: we do not find “reserches”. We double checked on the text to make sure we have the right spelling.